# CLUES: Collaborative Private-domain High-quality Data Selection for LLMs via Training Dynamics

**Wanru Zhao [1]\*, Hongxiang Fan[2,1], Shell Xu Hu [3], Bofan Chen [1], Nicholas D. Lane[1,4]**

[1] University of Cambridge, UK    [2] Imperial College London, UK
[3] Samsung AI Center Cambridge    [4] Flower Labs

## Abstract

Recent research has highlighted the importance of data quality in scaling large language models (LLMs). However, automated data quality control faces unique challenges in collaborative settings where sharing is not allowed directly between data silos. To tackle this issue, this paper proposes a novel data quality control technique based on the notion of data influence on the training dynamics of LLMs, that high quality data are more likely to have similar training dynamics to the anchor dataset. We then leverage the influence of the training dynamics to select high-quality data from different private domains, with centralized model updates on the server side in a collaborative training fashion by either model merging or federated learning. As for the data quality indicator, we compute the per-sample gradients with respect to the private data and the anchor dataset, and use the trace of the accumulated inner products as a measurement of data quality. In addition, we develop a quality control evaluation tailored for collaborative settings with heterogeneous medical domain data. Experiments show that training on the high-quality data selected by our method can often outperform other data selection methods for collaborative fine-tuning of LLMs, across diverse private domain datasets, in medical, multilingual and financial settings. Our code is released at CLUES.

## 1   Introduction

Large language models (LLMs) training has predominantly relied on the accumulation of vast datasets. Recent observations suggest that even a modest quantity of high-quality diverse data can significantly enhance the instruction following capacity of LLMs. Previously, data quality control relied heavily on manual selection processes [37, 36]. This approach, while being commonly used, rendered scalability challenges due to the substantial labor costs. Recent advancements have seen automated low-quality data filters [3], such as perplexity filters [29] and de-duplication filters [22]. However, their effectiveness in data quality control in more complex environments remains to be explored, where data are spread across silos and locations in different formats and difficult to find.

Collaborative training techniques, such as model merging [11] and federated learning [20], are common paradigms for addressing data-sharing constraints and GDPR [28] compliance. However, data quality control for private data is even more challenging if users are in charge of manually filtering data. We summarize here the two unique challenges: **(1) Quality Heterogeneity** Some clients may possess a higher proportion of low-quality data compared to others, thus we should not select data from all clients with a fixed selection ratio. **(2) Domain Heterogeneity** Different data silos may come from different vertical domains, for example, in the multilingual setting, different languages have different quality standards that are never unified.

---

*Corresponding Author: Wanru Zhao (`wz341@cam.ac.uk`)

38th Conference on Neural Information Processing Systems (NeurIPS 2024).

In this paper, we propose **CLUES** (**c**ollaborative **l**earning **u**nder **s**election), an automated high-quality data selection method for collaborative fine-tuning of Large Language Models (LLMs), showcasing notable performance improvements in mixed-quality data environments from different private domain data. In these domains, private LLM vendors are supposed to build their specialized applications based on open-source LLMs using their own private data, which represent specialized domains with significant private (e.g., patient records) and public data (e.g., scientific papers). By tracing the training dynamics of each training sample, we leverage public dataset to define an anchor dataset and compute the influence of each training sample on the anchor dataset, and set a global threshold to provide effective collaborative quality controls compared with traditional local data quality selection methods in the following aspects: *(1) General*: Our method is a general pipeline to improve the generalization performance for LLM fine-tuning. It has an interpretation in terms of bi-level optimization with inner optimization in the client side and outer optimization in the server side to minimize the loss on the anchor dataset. *(2) Collaborative*: Our method is a collaborative fine-tuning paradigm that can be seamlessly integrated into existing model merging and federated learning frameworks, where the modification occurs on the server side only to incorporate data selection. *(3) Scalable*: We only employ an approximation to solve the bi-level optimization, which makes it scalable to LLMs.

We evaluate our proposed method on medical, multilingual and financial Question Answering (QA) datasets, demonstrating significant improvements of up to 67.3% on challenging medical and financial QA datasets, highlighting the effectiveness of our proposed method. Through extensive analyses, we demonstrate the significant impact of leveraging training dynamics on the collaborative data quality control of LLMs.

## 2 Problem formulation: Collaborative Data Quality

### 2.1 Related Work

**Collaborative LLM Fine-Tuning Paradigms: Model Merging and Federated Learning**   Collaborative fine-tuning exhibits certain advantageous properties as a distributed machine learning paradigm by shifting the traditional model training process towards sharing model parameters instead of raw data. Participating clients train models using their own private datasets locally, and the updated model parameters are aggregated on the server. This preserves the privacy of the underlying data while collectively benefiting from the knowledge gained during the training process [20]. We focus on merging fine-tuned models that are optimized from the same pre-trained backbone. Different fine-tuned models initialized from the same pre-trained model effectively share a part of the optimization trajectory and can often be merged without accounting for permutation symmetry [40, 11, 17]. Therefore, merging fine-tuned models can improve performance on a single target task [13, 6], improve out-of-domain generalization [2, 1], create multitask models from different tasks [23], and other settings [23, 4].

One of the most significant challenges plaguing model merging and federated learning methods in previous research is the concern that the model parameters might interfere with each other during weighted averaging or other merging operations. This undesirable interaction could potentially lead to a merged model that performs worse than the individual models before merging. We argue that it can be tackled from the perspective of data attribution.

**Model merging.** let $f_\theta \in \mathcal{F}$ denote the language model and $D_k \in \mathcal{D}$ denote the training dataset on client $k$. Given the training datasets $D_k$, we can define a model merging operator $\mathcal{M}_K(\cdot; D_k, k \in K = \{1, \cdots, n\}) : \mathcal{F} \to \mathcal{F}$. The model merging process can be expressed as

$$f_{merging} = \mathcal{M}_K(f)$$

**Federated Averaging.** Based on the notation of model merging, the federated averaging process can be expressed as

$$f_{fed} = (\prod_{t=1}^{T} \mathcal{M}_{S_t(K)})(f)$$

where $T$ is the round number. $S_t(K)$ is the index set of clients participated in the training in round $t$.

**Data Attribution and Selection for LLMs**   The quality of the training data of a machine learning model can have a significant impact on its performance. One measure of data quality is the notion

of valuation, i.e., the degree to which a given training example affects the model and its predictive performance. Although data attribution is a well-known concept for researchers, the complexity behind large language models, coupled with their growing size, features, and datasets, has made quantification difficult. Recent methods include Perplexity Score, IFD [24], and DataInf [21], etc. More details are provided in Appendix E. However, those data attribution above have not been used in collaborative settings where each client has statistical heterogeneous and quality heterogeneous private-domain data. And previous data selection methods have not provide a way to determine the golden threshold to decide whether a training data sample should be kept or filter out.

**Training Dynamics** Previous works [38, 25, 35] that analyze training dynamics focus primarily on supervised learning and are largely model- and data-agnostic. Swayamdipta et al. [34] empirically demonstrated the influence of data by visually mapping individual training samples according to their impact on the correctness, confidence, and variability of a model.

## 2.2 Assumption and Objective: Collaborative High-quality Data Selection for LLMs

**Definition 1.1** (Data Quality on Specific Domain $k$). Given a model architecture $\theta$, a training configuration (optimizer, etc.), and a validation set $D_{val}$ in a specific domain $k$, the quality of training data $z$ is defined as follows: for $z_1, z_2 \in \mathcal{D}_{train}$, if $\mathcal{L}_{val}(\theta(z_1), D_{val}) < \mathcal{L}_{val}(\theta(z_2), D_{val})$, then the quality of $z_1$ is considered higher than that of $z_2$. Here, $\mathcal{L}_{val}$ denotes the validation loss. In other words, the lower the validation loss, the higher the data quality.

**Definition 1.2** (Data Quality in Collaborative Private Domains). Given a model architecture $\theta$, a training configuration (optimizer, etc.), and a validation set $D_{val} = \mathcal{D}_{val}^{(1)}, \mathcal{D}_{val}^{(2)}, \ldots, \mathcal{D}_{val}^{(K)}$ for all $K$ tasks, the quality of training data $z$ is defined based on the validation loss of the global model $\theta_{merged}$ on $D_{val}$. Specifically, for $z_1, z_2 \in \mathcal{D}_{train}^{(k)}$, if $\mathcal{L}_{val}(\theta_{merged}(z_1), D_{val}) < \mathcal{L}_{val}(\theta_{merged}(z_2), D_{val})$, then the quality of $z_1$ is considered higher than that of $z_2$. As in the single-domain case, lower validation loss indicates higher data quality.

**Remarks 1** (Impact of Low Quality Data in Collaborative Private Domains). We manually construct low-quality data samples on each client. We change the proportion of low-quality data from 0% to 100%. Higher scores indicate better performance. From Fig. 1, a larger portion of low-quality data results in higher validation loss, and more unstable and less effective training loss curve. Fig. 2 shows the performance drop when we change the proportion of low-quality data from 0% to 60%.

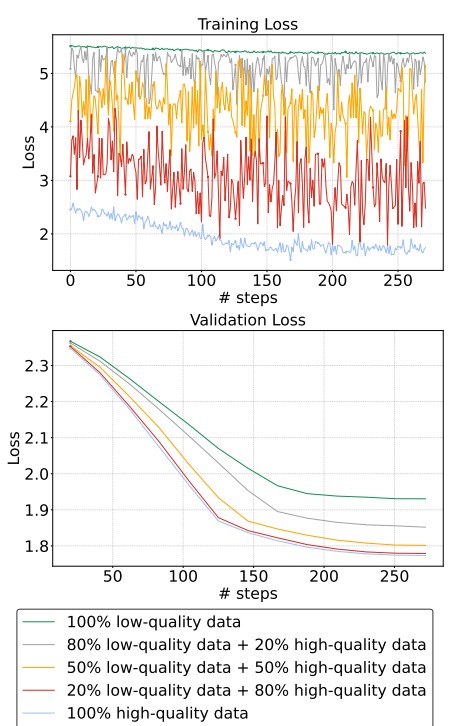

Figure 1: Validation loss and training loss.

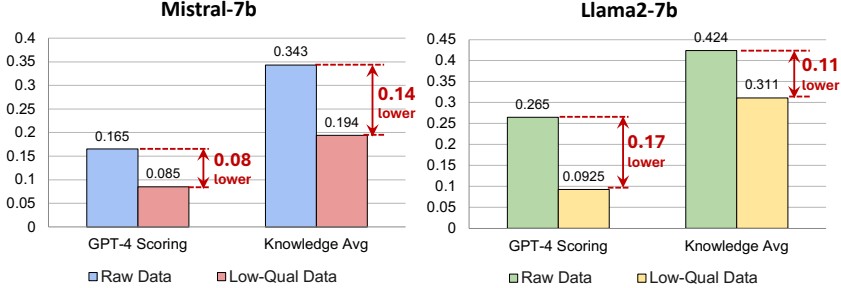

Figure 2: Performance drop on the performance of collaborative fine-tuning of LLMs when we change the proportion of low-quality data from 0% to 60%. Higher scores indicate better performance.

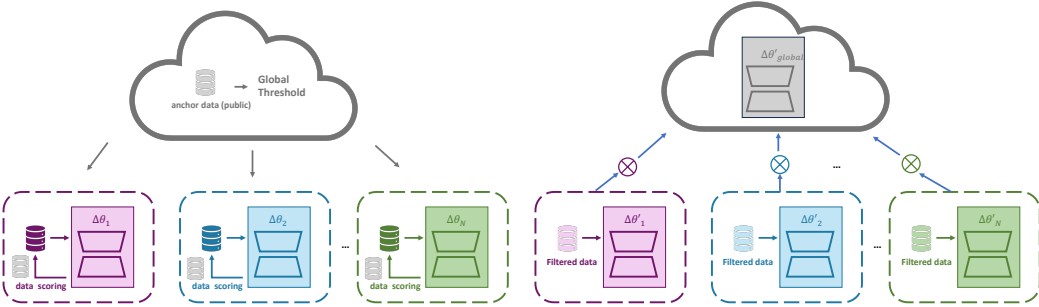

Step One: Local Training for Data Quality Scoring    Step Two: Collaborative Training with High-Quality Data

Figure 3: Overall workflow diagram consists of two phases: 1) Step One: client-side computes each sample's quality score with scoring functions using the public validation set and global model, then server-side calculates the score of a global threshold by anchor data 2) Step Two: clients filter data according to the global threshold and starts collaborative learning on selected high-quality data with adaptive weights on the model side.

**Remarks 2** (Enhancing Data Quality on Collaborative Private Domains). In the collaborative learning framework, the ratio and distribution of low-quality data are unknown a priori. Only the server has access to the global distribution of both high-quality and low-quality data, while individual clients cannot infer the global distribution from their local distributions due to statistical heterogeneity. The server can infer the distribution of high-quality data from public anchor data. Our objective is to select data points that most significantly reduce the validation loss of the global model, rather than optimizing for each local model independently. It is important to note that the scope of this study does not consider new models joining during training or continual learning paradigms.

## 3   Methodology: CLUES

### 3.1   Overview

In our workflow, each client performs local training using his own high-quality private data. We have a public validation set located on both the clients and the server, which consists of commonly recognized, high-quality public data. As illustrated in Figure 3, the overall workflow consists of two phases designed to achieve data quality control in the collaborative development of LLMs.

**Step One. Local Training for Data Quality Scoring** Local clients compute each sample's quality score via our training dynamics-based methods using the public validation set and their own fine-tuned model. The server determines a global threshold score, serving as a unified standard of data quality with only a very small amount of anchor data, and sends it to the clients.

**Step Two. Collaborative Learning with High-Quality Data** Each client then discards data samples that fall below the global threshold received, ensuring that only high-quality data verified by the unified standard are retained. The clients then utilize the high-quality filtered data sets $\mathcal{D}'_k$ (where $|\mathcal{D}'_k| \leq |\mathcal{D}_k|$) and the initial global model $\theta^0$ for collaborative learning. After local fine-tuning with the selected high-quality curation data, clients send their local LoRA adapter to the server. The server then aggregates the LoRA parameters of the individual models.

### 3.2   Step One: Training Dynamics-based Data Scoring

The idea behind our method is straightforward — trace the training process to capture changes in prediction as individual training examples are visited.

For each client, we have designed a data scoring step to calculate the score for each training data sample to measure its contribution to model prediction. Specifically, considering the training set of examples $\mathcal{D}_k = \{z_1, \ldots, z_K\}$ and a model $\theta$, we represent the validation set as $\mathcal{D}'_k = \{z'_1, \ldots, z'_K\}$. We measure the performance of a model using a loss function $\ell : \mathbb{R}^p \times Z \to \mathbb{R}$. The loss of the model noted by $\theta$ on an example $z$ is given by $\ell(\theta, z)$. We fine-tune the model by finding parameters $\theta$ that minimize the training loss $\sum_{i=1}^{K} \ell(\theta, z_i)$, through an iterative optimization procedure, such as

Stochastic Gradient Descent (SGD) or its variant, which utilizes one training example $z_t$ in iteration $t$, updating the parameter from $\theta_t$ to $\theta_{t+1}$:

$$\boldsymbol{\theta}^{t+1} - \boldsymbol{\theta}^t = -\eta_t \nabla \ell\left(\boldsymbol{z}; \boldsymbol{\theta}^t\right) \tag{1}$$

We trace the training process to capture changes in prediction as individual training examples are visited. The contribution of a particular training example $z$ on a given test example $z'$ is defined as the total reduction in loss on the test example $z'$ that is induced by the training process whenever the training example $z$ is utilized. We define the data quality of a particular training example $z$ as the sum of the contribution of the whole validation dataset.

The simplified expression for data quality is as follows:

$$S(z) = \sum_{z'} \sum_{t=1}^{T} \bar{\eta}_i \nabla \ell\left(\boldsymbol{z}', \boldsymbol{\theta}_t\right) \cdot \nabla \ell\left(\boldsymbol{z}, \boldsymbol{\theta}_t\right) \tag{2}$$

The per-sample gradients are calculated for each training sample from the checkpoint $t$ saved during the model training. LLMs are generally tuned using AdamW, which has a more complicated update formula involving the moving averages of the gradient moments.

For Adam,

$$\boldsymbol{\theta}^{t+1} - \boldsymbol{\theta}^t = -\eta_t \mathcal{L}\left(\boldsymbol{z}, \boldsymbol{\theta}^t\right), \mathcal{L}\left(\boldsymbol{z}, \boldsymbol{\theta}^t\right) \triangleq \frac{\boldsymbol{m}^{t+1}}{\sqrt{\boldsymbol{v}^{t+1}} + \epsilon} \tag{3}$$

For AdamW,

$$\boldsymbol{\theta}^{t+1} - \boldsymbol{\theta}^t = -\eta_t \mathcal{L}\left(\boldsymbol{z}, \boldsymbol{\theta}^t\right), \mathcal{L}\left(\boldsymbol{z}, \boldsymbol{\theta}^t\right) \triangleq \frac{\boldsymbol{m}^{t+1}}{\sqrt{\boldsymbol{v}^{t+1}} + \epsilon} + \lambda \boldsymbol{\theta}^t \tag{4}$$

Therefore, the training data quality score for LLMs is calculated using the following formula:

$$S(z) = \sum_{z'} \sum_{t=1}^{T} \bar{\eta}_i \mathcal{L}\left(\boldsymbol{z}', \boldsymbol{\theta}_t\right) \cdot \mathcal{L}\left(\boldsymbol{z}, \boldsymbol{\theta}_t\right) \tag{5}$$

The dot product of the loss gradients of the training example ($z$) and the test example ($z'$) is weighted by the learning rate ($\eta_i$) at different checkpoints and summed up, where we implemented applying point-wise loss gradients to disentangle the relative contributions of each training example. We use the output of the checkpoints from the learning algorithm to capture the training process. The higher the score $S(z)$, the higher the quality of the training sample $z$. We demonstrates an optimized training approach for collaborative learning of multiple models. By selecting high-quality training data for each local model, we select gradients that positively impact loss trajectories. These trimmed gradients accumulate, leading to an improved position in the weight space. Considering interference during our data selection (gradient selection) of $\Delta\theta'_1$ and $\Delta\theta'_2$, we reduce the interference of weight updates from different models. After parameter aggregation, the merged model $\Delta\theta_{merged}$ can be improved to an enhanced position in the weight space represented by $\Delta\theta_{targeted}$.

It is particularly well-suited for parameter-efficient fine-tuning techniques such as Low-Rank Adaptation (LoRA) [10], which involves freezing the pre-trained model weights and injecting trainable rank decomposition matrices into linear projects of the Transformer architecture. A neural network contains many dense layers that perform matrix multiplication. In the self-attention module, we denote the query projection matrices as $W_q$, the key projection matrices as $W_k$, the value projection matrices as $W_v$, the output project matrices as $W_o$. In principle, we can apply LoRA to any subset of weight matrices in a neural network to reduce the number of trainable parameters. In the Transformer architecture, there are four weight matrices in the self-attention module ($W_q, W_k, W_v, W_o$). In our implementation, we apply LoRA only to $W_q$ and $W_v$ in most experiments for simplicity.

One straightforward solution is to calculate the quality scores on all weight parameters of LoRA, but may be computationally infeasible when larger models with several millions of parameters are used. To address the memory bottleneck of calculating and saving gradients, we take gradients with respect to a given layer. We propose operating on the first layer of the model, which contains the

least cancelation effect, since the early layers encode *unique logit*. Therefore, we develop the idea of LoRA-based training-data influence in the context of gradient descent. Our proposed influence score is scalable due to the sparse nature of low-rank gradients and contains both low-level and high-level information since the gradient to the low-rank layer can capture both high-level and low-level information about the input sentence.

Note that the above gradient computation process is based on one single checkpoint and there is no parameter update throughout the process. Hence, for each training data point, we can perform this process in parallel, which can facilitate the computation.

### 3.3 Step Two: Global Standard with Anchor Data Scoring

On the server, we use a small set of public data (10 samples in our paper) as our anchor data and calculate the average score of these 10 data points as the global threshold. This establishes a unified standard for division between low- and high-quality data for heterogeneous clients, allowing for the further filtering of local data.

Then we merge the parameters of individual models with adaptive weights on different models. For model merging techniques, we implemented *Task Arithmetic* [12] on task weights, the LoRA matrices are involved in weighted sum. In task arithmetic, one first computes the task weights which is the difference between fine-tuned and base model weights, then calculates a weighted sum of these task weights. Here, the delta weights considered are the individual matrices $A$ and $B$ instead of their product $BA$. Consider two LoRA adapters $(A_1, B_1)$ and $(A_2, B_2)$ along with weights $w_1$ and $w_2$ for the weighted merging of these two adapters, then the merging happens as follows:

$$A_{\text{merged}} = \sqrt{w_1}A_1 + \sqrt{w_2}A_2 \tag{6}$$

$$B_{\text{merged}} = \sqrt{w_1}B_1 + \sqrt{w_2}B_2 \tag{7}$$

We also implement a more efficient method for merging LoRA adapters by eliminating redundant parameters: *TrIm, Elect, and Merge (TIES) [42]*. First, redundant parameters are trimmed, then conflicting signs are resolved into an aggregated vector, and finally, the parameters whose signs are the same as the aggregate sign are averaged. This method takes into account that some values (redundant and sign disagreement) can degrade performance in the merged model.

## 4 Experiments

Unlike traditional data quality selection methods for pre-trained models or traditional fine-tuning, in our collaborative setting, the training data from vertical domains is very sensitive and subject to strict restrictions regarding sharing and privacy. Therefore, we propose a new experimental setting using medical domain data for downstream tasks and evaluation for open-ended medical QA tasks, considering both quality heterogeneity and domain heterogeneity.

### 4.1 Experimental Setup

**Tasks and Datasets**   We conduct our evaluation on the open-ended question-answering (QA) tasks.

**(1) Medical QA:** PMC-LLama [41] and Medalpaca-flashcards [7] cover medical question-answering, rationale for reasoning, and conversational dialogues, comprising a total of 202M tokens. We use 16k samples in total, with 8k samples randomly sampled from PMC-LLama and Medalpaca-flashcards each. We uniformly partition the total samples into 20 clients in this task to demonstrate the effectiveness of CLUES in terms of the scalability of the clients, where the clients are IID subsets of the original distribution. For low-quality data, 3.2k samples (40% total data) are polluted with cutting, deletion, or substitution. These 40% low-quality data, together with the rest of the high-quality data, composites the mix-quality data set.

**(2) Multilingual QA:** MMedBench [33] is a medical muti-choice dataset of 6 different languages. It contains 45k samples for the trainset and 8,518 samples for the testset. Each question is accompanied by a right answer and high-quality rationale. We use 6312 samples randomly sampled from MMed-Bench and 1052 samples per language for each of the 6 clients. For the low-quality data, a certain ratio is either polluted with random noise.

**(3) Financial QA:** To demonstrate the generalizability of our proposed method across various domains, we also include FiQA [5], part of the training corpus of FinGPT [43], which consists 17.1k financial open Question-Answering instructions. We randomly sample 2000 data samples for each of the 4 clients from FiQA dataset, and pollute each of them with a low-quality data ratio of 80%, 20%, 10%, and 50% respectively.

Note that for all tasks, the anchor data and validation dataset used in our proposed method are selected as a held-out high-quality dataset from the same data source.

**Models** We use LLama2-7b [37] and Mistral-7b [14] as our pre-trained models, and fine-tune them with Low-Rank Adaptation (LoRA) [10] on each of the client side. As for the model merging technique, in our main experiments, we use TIES merging. We also compare it with Task Arithmetic in our ablation studies.

**Baselines** The *Oracle* shows the results that train only on the remaining high quality data in the mixed-quality dataset, serving as the theoretical upper bound. We implement the three baselines: existing methods mentioned in 2.1: Perplexity score, IFD [24], and DataInf [21] independently on each client.

**Evaluation metrics** The evaluations focus on two main aspects: **(1) Question-Answering capabilities**, assessed by GPT-4 [30] scoring within the test set splited from the same sources of the training dataset. In the medical QA and multilingual QA tasks, 200 samples are randomly selected from the medical dataset to serve as the test set. We evaluate the models that need to be compared on the test set to generate responses respectively. Then we use the OpenAI GPT-4 model API to assign scores to their responses. Each response of is rated by the judge on a scale from 0 to 1, reflecting how well the answer aligns with the ground truth. In our financial QA task, GPT-4 rate the responses of the fine-tuned model on our data set on a scale of 1 to 10, reflecting criteria including relevance, precision and fluency. To address potential positional bias, we send our response along with the benchmark output to GPT-4 twice, with different orders. We then calculate the average of these scores as the final performance score. **(2) Knowledge acquisition**, measured by average accuracy of responses to multiple-choice questions in the MMLU clinical topics [8, 9], MedMCQA [31], PubMedQA [16], and USMLE [15] datasets. Although the goal of private domain fine-tuning is not to increase knowledge, there shouldn't be too much knowledge forgetting during this process. **(3) Data selection correctness** Precision, Recall, F-1 Score, and Accuracy are widely-used evaluation metrics that provide complementary insights into the model's effectiveness from the data selection perspective. In our case, positive instances represent high-quality data, while negative instances represent low-quality data. Precision quantifies the proportion of correctly identified positive instances among all instances predicted as positive, while Recall measures the proportion of correctly identified positive instances among all actual positive instances in the dataset. The F1 Score offers a balanced measure of Precision and Recall, while Accuracy reflects the overall correctness of our data selection (based on our data scoring and threshold determining method) across all classes.

## 4.2 Main Results

Based on the low-quality dataset setup, we evaluate our data-quality control pipeline in collaborative LLM fine-tuning in both federated (communication round $cr = 300$) and model merging (communication round $cr = 1$) settings. Note that in federated learning, the server and clients need to intensively communicate the model updates during model training. We implement the three baseline methods described in the Section 2.1 to calculate scores for each training data, and set the unified scoring standard using corresponding scoring functions with anchor data.

We demonstrate the performance of data quality control methods in collaborative settings in the medical QA task (Tab. 1) and Multilingual QA task (Tab. 2).

**Federated Learning v.s. Model Merging** Firstly, for both pre-trained models and tasks, with other settings remaining the same, model merging performs better than federated learning. This indicates that loose communication between the local model and the server, compared to frequent communication, might lead to better generalization. Additionally, the performance boost with selected data in the federated setting is larger than in the model merging setting. This might be because during federated learning, we calculate the data score based on the global model (instead of the local model

Table 1: Data selection performance in federated setting on MedicalQA. We **bold** the highest performance and underline the second highest performance for each row.

| | Mistral-7b | | Llama2-7b | |
|---|---|---|---|---|
| Evaluation Metric | GPT-4 Scoring | Knowledge Avg | GPT-4 Scoring | Knowledge Avg |
| Mix-qual Data | 0.085 | 0.194 | 0.0952 | 0.311 |
| Oracle | 0.160 | 0.233 | 0.099 | **0.440** |
| PPL | 0.079 | **0.346** | 0.045 | 0.362 |
| IFD [19] | 0.087 | 0.287 | 0.050 | 0.346 |
| DataInf [24] | 0.093 | 0.106 | 0.103 | 0.335 |
| CLUES (ours) | **0.161** (100.6%) | 0.309 (132%) | **0.210** (212.1%) | 0.356 (80.9%) |

Table 2: Data selection performance on MMedBench. We **bold** the highest performance and underline the second highest performance for each row.

| | Mistral-7b | | Llama2-7b | |
|---|---|---|---|---|
| Setting | Federated | Model Merging | Federated | Model Merging |
| Mix-qual Data | 0.420 | 0.515 | 0.440 | 0.485 |
| Oracle | **0.451** | **0.530** | 0.449 | **0.490** |
| CLUES (ours) | 0.435 (96.5%) | 0.525 (99.1%) | **0.477** (106.2%) | 0.487 (99.4%) |

in the model merging setting) at each timestamp, which can better trace and regularize the training trajectory to the optimal location.

**Data Selection Performance**   In both federated and model merging settings, our data selection can achieve over 96% and over 91% of the theoretical upper bound performance, respectively. Our method outperforms the other local data selection baselines under the GPT4 Scoring metrics. Compared to the other methods which cause severe forgetting during instruction tuning, the performance of our method on the Knowledge-based benchmark remains within an acceptable range. This shows that our methods are able to improve domain-specific tasks without forgetting knowledge injected during pretraining.

## 5   Analysis

### 5.1   Qualitative Analysis

We performed a qualitative analysis by manually comparing the outputs generated by models fine-tuned on our selected high-quality data versus the original low-quality data. This comparison (Tab. 7 and Tab. 8) provides insights into the tangible improvements in model performance and output quality.

### 5.2   Varying Levels of Low-Quality Data

To evaluate the robustness of our data selection method under different data quality conditions, we conducted a series of experiments with varying proportions of low-quality data. We maintained a consistent proportion of low-quality data across all clients for each experiment, ranging from 0% to 100%, including pollution levels 20%, 50%, and 80%.

Fig. 4 presents the performance of models trained with and without our data selection method across these different proportions. The results demonstrate that our method effectively enhances data quality across all scenarios with GPT-4 scoring. And in terms of accuracy of the data selection, our method consistently selected over 99% of the high-quality data across different proportions of low-quality data. Additionally, to understand the adaptability of our global threshold, we analyzed how the global threshold changes with different proportions of low-quality data. Fig. 4 illustrates that our global threshold adjusts across varying levels of data quality.

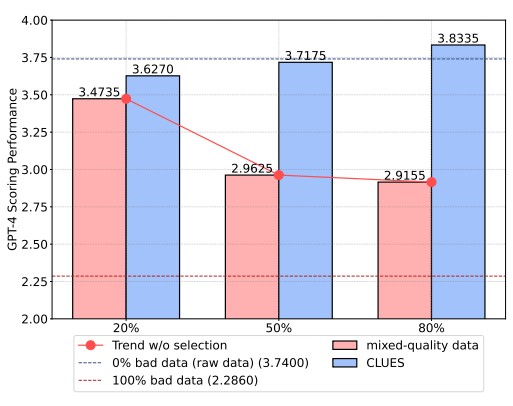
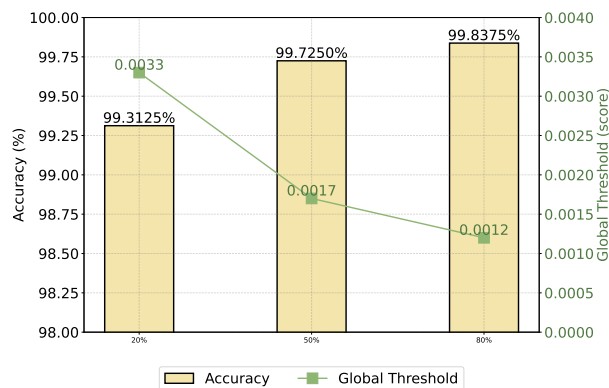

(a) GPT-4 Scoring Performance in different proportions of low-quality data.

(b) Selection accuracy and global threshold (score) in different proportion of low-quality data.

Figure 4: Experimental results for different levels of low-quality data

## 5.3 Quality Heterogeneity

To provide a more comprehensive analysis, in addition to the experiments in the *domain heterogeneity* setting shown above, we conducted additional experiments in a *quality heterogeneity* setting using the FiQA dataset, which focuses on the answer of financial questions. Specifically, we randomly polluted 80%, 20%, 10%, and 50% of the training set for each of the four clients, respectively. The findings demonstrate that our method significantly enhances the quality of the data even when clients have different proportions of low-quality data.

**Varying Merging Techniques** Fig. 5 demonstrates that different weighted merging or aggregation techniques lead to varying performance. Notably, the performance of our data selection method with the *Linear Merging* technique does not even reach the performance of low-quality data with *TIES Merging* technique, highlighting the significant impact of weighted merging techniques on overall performance. Furthermore, we experimented with different merging techniques on the FiQA dataset, demonstrating the importance of weighted merging, shown in Fig. 5.

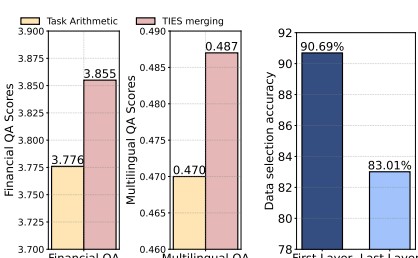

Figure 5: Left: Comparison of different merging techniques. Right: First layer v.s. last layer for low-rank tracing gradient.

**Layer Selection for Low-Rank Tracing Gradient** In terms of layer selection, we evaluated both the last layer and the *token embeddings*. We show that layer selection distorts the score (the inner product of two gradients). In our ablation study, we observe that since the activation connected to the last layer of weights contains *shared logic*, the data influenced calculated through the last layer weights are prone to a *cancellation effect*, where the data influence of different examples has a large magnitude that contradicts each other. The cancelation effect lowers the power of the influence score, and deleting influential examples according to this measure often does not change the model's behavior by much. From Fig. 5, we show that the first layer has a less severe cancelation effect than the last layer.

**Unified Scoring with Anchor Data** We conducted an ablation study on our global threshold to further validate our approach. Tab. 3 illustrates the advantage of using a global threshold determined by our anchor data for data selection in this heterogeneous setting, compared to selection based on average ratio or pre-determined scores. These results demonstrate that our approach successfully balances the identification of positive cases with the minimization of false positives, offering a robust and superior solution.

Table 3: Data selection performance on FiQA. We **bold** the highest performance for each row.

|  | Precision | Recall | F1 Score | Accuracy |
|---|---|---|---|---|
| Select by ratio | 79.17% | 79.17% | 79.17% | 75.00% |
| Select by a pre-determined score | 92.77% | 99.13% | 95.84% | 95.00% |
| Select by global threshold **(Ours)** | **97.44%** | **99.38%** | **98.39%** | **97.91%** |

## 6  Discussion and Conclusion

Collaborative model development, including model merging and federated averaging, would benefit from different kinds of high-quality data, and for each of them, the definition of quality is slightly different. In this paper, we establish a data quality control pipeline for collaborative fine-tuning of LLMs, avoiding directly sharing any private data. Our experiments show that the selected high-quality data ensures an effective and reliable learning process, leading to improved model performance.

To the best of our knowledge, we are the first to propose a data selection method for large language models in a collaborative setting, while previous work has mainly focused on traditional centralized settings. We bring up the insights to view federated learning and model merging within the same framework, incorporate different experimental setups and unify federated learning and model merging methods, making it universally applicable. Additionally, our method performs well on generation datasets and takes into account scenarios with bad data, while previous work has not considered downstream domain-specific generation tasks for large language models. Our method does not require repeated training.

**Societal impact**   Our work builds large language models that make it possible to create a collaborative instead of a monolithic ecosystem from open-source models while preserving the privacy of users' own data. The constant progress being made in machine learning needs to extend across borders if we are to democratize ML in developing countries. Adapting state-of-the-art (SOTA) methods to resource-constrained environments such as developing countries can be challenging in practice, pushing open source and inclusion.

**Limitations and future work**   Our data quality control methods are based on the assumption that all the local models share the same model architectures. It is easy to achieve when our fine-tuning is based on the LoRA adapter. However, it may be worth extending it to adapt to different local model architectures, for example, different low ranks. Future work may explore the intrinsic relation between data selection and the model parameters and how our data selection methods can help reduce the interference of parameter vectors from different models.

## Acknowledgement

The authors would like to thank Colin Raffel, Haokun Liu, Meghdad Kurmanji and Stefanos Laskaridis for useful discussions and feedback. This research was supported by the European Research Council via the REDIAL project and the Royal Academy of Engineering via the DANTE.

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

# A  Data Attribution

*Perplexity (PPL)* serves as a fundamental metric in language modeling to measure the model's ability to predict text sequences accurately. Mathematically, it is defined as the exponential of the average negative log-likelihood: $\text{PPL} = \exp(-\frac{1}{N}\sum_{i=1}^{N}\log P(w_i|w_1,\ldots,w_{i-1}))$, where $N$ is the number of tokens in the sequence and $P(w_i|w_1,\ldots,w_{i-1})$ represents the probability the model assigns to token $w_i$ given its preceding context. A lower perplexity score indicates better model performance, as it suggests that the model assigns higher probabilities to the correct tokens in the sequence, effectively measuring how "surprised" the model is by new text.

*Instruction Following Difficulty (IFD)* [24] provides a quantitative metric for evaluating the difficulty of instruction-following tasks in language models. This score is calculated as the ratio between two key measurements: the Conditioned Answer Score $s_\theta(A|Q)$ and the Direct Answer Score $s_\theta(A)$: $\text{IFD}_\theta(Q,A) = \frac{s_\theta(A|Q)}{s_\theta(A)}$, where $s_\theta(A|Q)$ measures the model's ability to generate responses with instructional context, and $s_\theta(A)$ evaluates the model's capability to generate answers in isolation. The Direct Answer Score is computed as $s_\theta(A) = -\frac{1}{N}\sum_{i=1}^{N}\log P(w_i^A|w_1^A,\ldots,w_{i-1}^A;\theta)$. This metric quantifies the extent to which instructions aid in response generation, where a higher IFD score suggests that the given instruction provides limited useful context for the model's response generation, indicating greater difficulty in following the instruction.

*Influence Functions [19]:*  DataInf represents an efficient algorithm for computing influence functions, distinguished by its closed-form expression that reduces computational and memory complexity compared to existing methods.  The algorithm approximates the inverse Hessian calculation $(G_l(\theta^*) + \lambda_l I_{d_l})^{-1}$ through the key transformation: $\frac{1}{n}\sum_{i=1}^{n}\left(\nabla_{\theta_l}\ell_i\nabla_{\theta_l}\ell_i^T + \lambda_l I_{d_l}\right)^{-1} \approx \frac{1}{n}\sum_{i=1}^{n}\left(I_{d_l} - \frac{\nabla_{\theta_l}\ell_i\nabla_{\theta_l}\ell_i^T}{\lambda_l+\nabla_{\theta_l}\ell_i^T\nabla_{\theta_l}\ell_i}\right)$, where the Sherman-Morrison formula enables a closed-form solution.  The influence function is computed as $\mathcal{I}_{\text{DataInf}}(x_k,y_k) = \sum_{l=1}^{L}\frac{1}{\lambda_l}\left(\frac{1}{n}\sum_{i=1}^{n}\frac{L_{l,i}}{\lambda_l+L_{l,ii}}L_{l,ik} - L_{l,k}\right)$.

# B  Priliminary Results on Low-quality Data

Table 4: Preliminary results on MedicalQA.

| | Mistral-7b | | Llama2-7b | |
|---|---|---|---|---|
| Evaluation Metric | GPT-4 Scoring | KnowledgeAvg | GPT-4 Scoring | Knowledge Avg |
| Raw Data | 0.165 | 0.343 | 0.265 | 0.424 |
| Mix-qual Data | 0.085 (↓48.5%) | 0.194 (↓43.4%) | 0.0925 (↓65.1%) | 0.311 (↓26.7%) |

Table 5: Preliminary results on MMedBench.

| | Mistral-7b | | Llama2-7b | |
|---|---|---|---|---|
| Setting | Federated | Model Merging | Federated | Model Merging |
| Raw Data | 0.455 | 0.540 | 0.450 | 0.505 |
| Mix-qual Data | 0.420 (↓7.69%) | 0.515 (↓6.48%) | 0.440 (↓2.22%) | 0.485 (↓3.96%) |

# C  Detailed Method Description

**Stage 1 (On each client)** Local fine-tuning with low-quality data; save model checkpoints.
**Stage 2 (On each client)** Calculate gradients, compute scores for each training sample, send scores to the server.
**Stage 3 (On the server)** Calculate gradients, compute scores of anchor data, determine the global threshold using anchor data scores and client scores.
**Stage 4 (On each client)** Select data with scores not lower than the global threshold.
**Stage 5 (On each client)** Local fine-tuning with high-quality data, then send model parameters to the server.
**Stage 6 (On the server)** Merge client model parameters to obtain the final global model.

---

**Algorithm 1** Our data selection method for collaborative fine-tuning

---

***Initialization*** Initial global model: $\theta^0$; Training datasets (private):$D_{train} = \left\{ \mathcal{D}_{train}^{(1)}, \mathcal{D}_{train}^{(2)}, \dots, \mathcal{D}_{train}^{(K)} \right\}, \mathcal{D}_{train}^{(k)} = \{z_1^{(k)}, \dots, z_n^{(k)}\};$

Validation datasets (public): $D_{val} = \left\{ D_{val}^{(1)}, \mathcal{D}_{val}^{(2)}, \dots, \mathcal{D}_{val}^{(K)} \right\}, \mathcal{D}_{val}^{(k)} = \{z_1'^{(k)}, \dots, z_m'^{(k)}\};$

Anchor data (public): $D_{anc} = \left\{ \mathcal{D}_{anc}^{(1)}, \mathcal{D}_{anc}^{(2)}, \dots, \mathcal{D}_{anc}^{(K)} \right\}, \mathcal{D}_{anc}^{(k)} = \{z_1^{*(k)}, \dots, z_v^{*(k)}\};$

---

***On the Client*** $k$

Train model $\theta_k$ on $\mathcal{D}_{train}^{(k)}$  ▷ *Training with Mixed Quality data*

**for** each training sample $z_i \in \mathcal{D}_{train}^{(k)}$ **do**

    Calculate $\nabla\ell(z', \boldsymbol{\theta}_t)$

    **for** each checkpoint $t$ in $T$ **do**

        **for** each validation sample $z_i' \in \mathcal{D}_{val}^{(k)}$ **do**

            Calculate $\nabla\ell(z_i', \boldsymbol{\theta}_t)$

        **end for**

    **end for**

    $S(z) = \sum_{z'} \sum_{t=1}^{T} \bar{\eta}_i \nabla\ell(z', \boldsymbol{\theta}_t) \cdot \nabla\ell(z, \boldsymbol{\theta}_t)$  ▷ *Data Scoring*

**end for**

$\mathcal{D'}_{train}^{(k)} = \left\{ z_i \in D_{train}^{(k)}, z_i \geq \tau \right\}$  ▷ *Select training data with scores above the threshold*

Train model $\theta_k'$ on $\mathcal{D'}_{train}^{(k)}$  ▷ *Training with High-Quality Data*

Send updated $\theta_k'$ to server

---

***On the Server***

**for** each anchor data $z_i \in \mathcal{D}_{anc}^{(k)}$ **do**

    Calculate $\nabla\ell(z, \boldsymbol{\theta}_t)$

    **for** each checkpoint $t$ in $T$ **do**

        **for** each validation sample $z_i' \in \mathcal{D}_{val}^{(k)}$ **do**

            Calculate $\nabla\ell(z_i', \boldsymbol{\theta}_t)$

        **end for**

    **end for**

    $S(z) = \sum_{t=1}^{T} \bar{\eta}_i \nabla\ell(z', \boldsymbol{\theta}_t) \cdot \nabla\ell(z, \boldsymbol{\theta}_t)$  ▷ *Anchor data score*

**end for**

Determine the global threshold $\tau$ with anchor data $D_{anc}$

Send $\tau$ to each clients for training with High-quality data

Aggregate client updates: $\theta' = \sum_{k=1}^{K} \frac{|\mathcal{D}_k|}{\sum_{k=1}^{K} |\mathcal{D}_k|} \theta_k'$  ▷ *Model Merging or Aggregation*

---

## D    Complexity of Data Scoring

The overall compute complexity, where $N$ is number of checkpoints, and $d$ is gradient dimension.

$$\mathcal{O}\left(N \cdot |\mathcal{D}| \cdot |\mathcal{D}_{\text{val}}| \cdot d\right)$$

The overall storage complexity:

$$\mathcal{O}(|\mathcal{D}| \cdot N \cdot d + |D_{val}| \cdot N \cdot d)$$

## E    Implementation Details

We use Hugging Face's transformers library [39] and PEFT library [27] for loading pre-trained models and LoRA configurations. For our training and evaluation, we use PyTorch as the underlying auto-differentiation framework [32]. We use the AdamW optimizer [26, 18] for all experiments. All the experiments are conducted on the machines with the same hardware configuration using

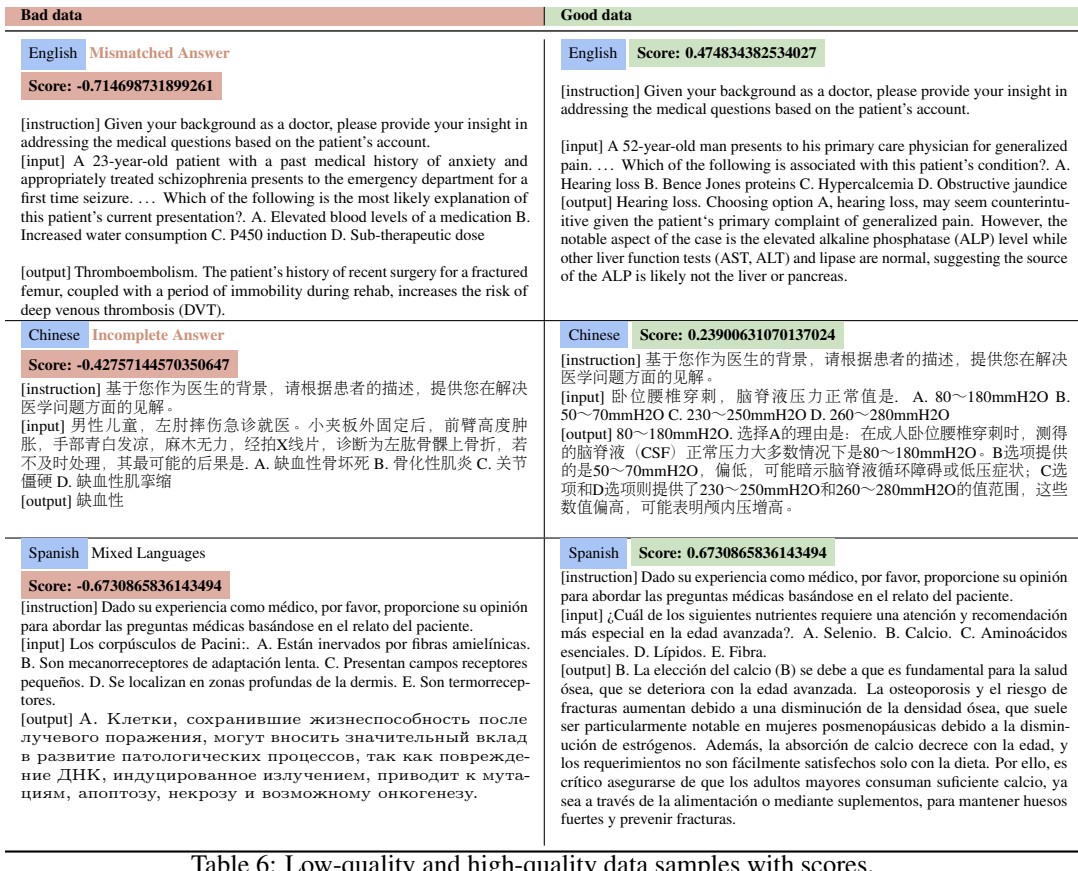

| Bad data | Good data |
|---|---|
| **English** **Mismatched Answer**
**Score: -0.714698731899261**

[instruction] Given your background as a doctor, please provide your insight in addressing the medical questions based on the patient's account.
[input] A 23-year-old patient with a past medical history of anxiety and appropriately treated schizophrenia presents to the emergency department for a first time seizure. . . . Which of the following is the most likely explanation of this patient's current presentation?. A. Elevated blood levels of a medication B. Increased water consumption C. P450 induction D. Sub-therapeutic dose

[output] Thromboembolism. The patient's history of recent surgery for a fractured femur, coupled with a period of immobility during rehab, increases the risk of deep venous thrombosis (DVT). | **English** **Score: 0.474834382534027**

[instruction] Given your background as a doctor, please provide your insight in addressing the medical questions based on the patient's account.

[input] A 52-year-old man presents to his primary care physician for generalized pain. . . . Which of the following is associated with this patient's condition?. A. Hearing loss B. Bence Jones proteins C. Hypercalcemia D. Obstructive jaundice
[output] Hearing loss. Choosing option A, hearing loss, may seem counterintuitive given the patient's primary complaint of generalized pain. However, the notable aspect of the case is the elevated alkaline phosphatase (ALP) level while other liver function tests (AST, ALT) and lipase are normal, suggesting the source of the ALP is likely not the liver or pancreas. |
| **Chinese** **Incomplete Answer**
**Score: -0.42757144570350647**
[instruction] 基于您作为医生的背景，请根据患者的描述，提供您在解决医学问题方面的见解。
[input] 男性儿童，左肘摔伤急诊就医。小夹板外固定后，前臂高度肿胀，手部青白发凉，麻木无力，经拍X线片，诊断为左肱骨髁上骨折，若不及时处理，其最可能的后果是. A. 缺血性骨坏死 B. 骨化性肌炎 C. 关节僵硬 D. 缺血性肌挛缩
[output] 缺血性 | **Chinese** **Score: 0.23900631070137024**
[instruction] 基于您作为医生的背景，请根据患者的描述，提供您在解决医学问题方面的见解。
[input] 卧位腰椎穿刺，脑脊液压力正常值是. A. 80～180mmH2O B. 50～70mmH2O C. 230～250mmH2O D. 260～280mmH2O
[output] 80～180mmH2O. 选择A的理由是：在成人卧位腰椎穿刺时，测得的脑脊液（CSF）正常压力大多数情况下是80～180mmH2O。B选项提供的是50～70mmH2O，偏低，可能暗示脑脊液循环障碍或低压症状；C选项和D选项则提供了230～250mmH2O和260～280mmH2O的值范围，这些数值偏高，可能表明颅内压增高。 |
| **Spanish** Mixed Languages
**Score: -0.6730865836143494**
[instruction] Dado su experiencia como médico, por favor, proporcione su opinión para abordar las preguntas médicas basándose en el relato del paciente.
[input] Los corpúsculos de Pacini:. A. Están inervados por fibras amielínicas. B. Son mecanorreceptores de adaptación lenta. C. Presentan campos receptores pequeños. D. Se localizan en zonas profundas de la dermis. E. Son termorreceptores.
[output] A. Клетки, сохранившие жизнеспособность после лучевого поражения, могут вносить значительный вклад в развитие патологических процессов, так как повреждение ДНК, индуцированное излучением, приводит к мутациям, апоптозу, некрозу и возможному онкогенезу. | **Spanish** **Score: 0.6730865836143494**
[instruction] Dado su experiencia como médico, por favor, proporcione su opinión para abordar las preguntas médicas basándose en el relato del paciente.
[input] ¿Cuál de los siguientes nutrientes requiere una atención y recomendación más especial en la edad avanzada?. A. Selenio. B. Calcio. C. Aminoácidos esenciales. D. Lípidos. E. Fibra.
[output] B. La elección del calcio (B) se debe a que es fundamental para la salud ósea, que se deteriora con la edad avanzada. La osteoporosis y el riesgo de fracturas aumentan debido a una disminución de la densidad ósea, que suele ser particularmente notable en mujeres posmenopáusicas debido a la disminución de estrógenos. Además, la absorción de calcio decrece con la edad, y los requerimientos no son fácilmente satisfechos solo con la dieta. Por ello, es crítico asegurarse de que los adultos mayores consuman suficiente calcio, ya sea a través de la alimentación o mediante suplementos, para mantener huesos fuertes y prevenir fracturas. |

Table 6: Low-quality and high-quality data samples with scores.

NVIDIA A40 and A6000. In the experiments of Medical QA, we consider 300 communication rounds, $n = 20$ clients, 10 local steps of model training. In the experiments of Multilingual Medical QA, we distrubute 6 languages on 6 clients, with each clients 1052 dataset. In all experiments, we use 8 bit quantization, set $batch\_size = 16$, and LoRA rank to 16 (all the participating LoRA adapters have same rank), and we do 3 epochs for local training.

## F   Additional Experimental Results

| Questions | Ground Truth Response | Model Response (w/o data selection) | Model Response (w/ our data selection) |
|---|---|---|---|
| 初产妇，23岁，规律宫缩10小时，持续观察2小时，宫口由6cm开大至7cm，胎头+1，胎心140次/分。恰当的处置应为. A.严密观察产程进展 B. 静脉滴注缩宫素 C. 立即行人工破膜 D. 立即行剖宫手术 | 严密观察产程进展. 选择A（严密观察产程进展）的原因是因为初产妇宫口开大至7cm，这表明产程正在缓慢进展，但并无明显异常。胎头已下降至+1位置，这是正常的胎位下降过程。胎心率140次/分属于正常范围，说明胎儿目前处于安全状态。 | 选择D. 立即行剖宫手术. 初产妇的宫缩是一种常见的产前症状，宫缩的规律性和持续时间可以反映出妊娠的进展，但它并不代表产妇的产前状况。选项A. 严密观察产程进展，是一种产前监测的方法，但不能代表产妇的产前状况。选项B. 静脉滴注缩宫素，是一种产前治疗方法，但不能代表产妇的产前状况。 | 恰当的处置应为A.严密观察产程进展.初产妇的宫缩是一种常见的产前症状，它可能表示妊娠进展正常，也可能是产前妊娠综合征的一种表现。严密观察产程进展是对初产妇的产前监测的重要组成部分，它可以帮助医生确定产前妊娠综合征的发生情况，并采取适当的治疗措施。因此，选项A.严密观察产程进展是最合适的处置选项。 |
| What type of injury to the arm/elbow most often leads to supracondylar fractures? | Supracondylar fractures most often occur after hyperextension injuries of the arm/elbow. | 1000 Hg is the pressure required to rupture an average aortic aortic | Supracondylar fractures are a type of fracture that occurs above the elbow joint, and they are most commonly caused by a fall onto an outstretched arm. |

Table 7: Samples of the output of merged models on multilingual MMedBench dataset.

| Questions | Model Response (w/o data selection) | Model Response (w/ our data selection) |
|---|---|---|
| How does a 2 year treasury note work? | It is a 2 year treasury note. | Treasury notes are a short-term government security that matures in two years. They are issued by the U.S. Treasury Department to help ìnance the nation's deìcit. |
| What causes a stock to drop in price? | Investors are afraid to buy stocks because they are afraid the stock will drop in price. | If the stock price is falling, it's because there's a lot of supply, but not enough demand. If there's a lot of demand, but not enough supply, the price will rise. |

Table 8: Samples of the output of merged models on FiQA dataset.

