# OpenReview forum: "CLUES: Collaborative Private-domain High-quality Data Selection for LLMs via Training Dynamics"
_NeurIPS.cc/2024/Conference — NeurIPS 2024 poster_

### Official Review · Reviewer_V7za · 2024-07-12

**Soundness:** 3
**Presentation:** 4
**Contribution:** 3
**Rating:** 4
**Confidence:** 4

**Summary:**

The paper proposes a novel data quality control technique to enhance the quality of data from different private domains in collaborative training settings, such as model merging. The technique scores each training sample by tracing gradients of low-rank adapters and filters out low-quality data locally. The model parameters are then adaptively merged based on individual training. The method is evaluated on heterogeneous medical domain data, demonstrating that training on high-quality data selected by this method outperforms other data selection methods across diverse private domain datasets. The paper claims performance improvements and offers a general, efficient way to identify high-quality data for large language models.

**Strengths:**

s1. The use of training dynamics to score and filter data for collaborative fine-tuning is a good idea.

s2. The method provides a general pipeline applicable to various models without the need for task-specific adjustments.

s3. The structure of the article is complete and easy to understand

**Weaknesses:**

w1. The paper's evaluation is confined to medical datasets. To demonstrate the generalizability and robustness of the proposed method, experiments should be conducted on a wider range of datasets from different domains.

w2. The paper generates 40% of the data as low-quality, which may not accurately reflect real-world scenarios. It would be beneficial to experiment with varying proportions of low-quality data to understand the method's effectiveness across different levels of data quality.

w3. The method uses only a small amount of public data (10 samples in the paper) as anchor data to calculate the global threshold. The reliability of selecting just 10 samples is questionable, and the basis for this choice should be clarified. A more rigorous justification or experimentation with different numbers of anchor samples could strengthen the method's credibility.

w4. Table 1 in the paper shows that the proposed method performs better than the Oracle method, which uses only high-quality data. The paper does not provide an analysis or explanation for this superior performance. Understanding and explaining why this method outperforms the Oracle method could provide valuable insights and strengthen the overall argument.

**Questions:**

Q1. Please explain how the proposed method performs on datasets from different domains beyond the medical field. Have you considered conducting experiments with datasets from various other domains to validate the generalizability of your approach? (w1)

2. Please clarify if 40% is representative of real-world scenarios. Would you consider testing with different proportions of low-quality data to assess the robustness and effectiveness of your method under varying conditions? (w2)

3. Have you tested the impact of using different numbers of anchor samples on the performance and reliability of the method?(w3)

4. Please explain why our method performs better than Oracle and why the other three data evaluation methods perform so poorly in Table 1. (w4)

**Limitations:**

1. The author's work is similar to federated learning. Should it be compared with the recent federated learning?

---

> ### Author Rebuttal · Authors · 2024-08-07
>
> **W1 & Q1**:
> > Please explain how the proposed method performs on datasets from different domains beyond the medical field.
>
> As mentioned in s1, our method by nature can be adapted to various models ‘without the need for task-specific adjustments’. We conduct comprehensive evaluations on different medical and healthcare dataset which covers different domains in medicine. Our assumptions are not limited to any specific domain, and our selection of anchor data is not domain-specific. We chose medical data because of the availability of high-quality public datasets in this field.
>
> We are running the experiments on the financial dataset: [FiQA](https://huggingface.co/datasets/LLukas22/fiqa). We randomly sampled 2000 data samples for each of the 4 clients from FiQA dataset, and polluted each of them with a low-quality data ratio of 80%, 20%, 10%, and 50% respectively. Once we get the result, we’ll update our response.
>
> **W2 & Q2**
> > Experiment with varying proportions of low-quality data to understand the method's effectiveness across different levels of data quality.
> We have been running the experiments on different proportions of low-quality data: 80% and 100%. Once we get the result, we’ll update our response.
>
> We reported the estimated running time of our collaborative fine-tuning pipeline (for each proportion of low-quality data on a certain dataset on A40). Due to the time limits, we are still running the experiments and expected to have additional experimental results within the discussion period.
>
> |Step	|Estimated Time|
> |----|-----|
> |Collaborative finetuning with low-quality data|	9 hours|
> |Calculate gradients|	3 hours|
> |Compute scores and select|5 hours|
> |Collaborative finetuning with selected data|	9 hours|
> |Prediction on test set|	8 hours|
> |GPT-4 evaluation|	0.5 hour|
>
> **W3 & Q3**
> > Impact of using different numbers of anchor samples on the performance and reliability of the method
>
> We show the ablation study of the anchor data selection and validation set selection in Figure 2 in our PDF, which demonstrates the robustness and unbiased selection in our experiments. Fig 2 shows that the threshold keeps stable with the increase of the number of anchor data, thus demonstrating the robustness of the anchor data selection.
>
> I would like to also highlight that the number of anchor data will not affect the order of the selected high-quality data, the only impact is the number of data selected.
>
> **W4 & Q4**
> > Why our method performs better than Oracle?
>
> In our experimental setup, the global threshold is $0.48$, and the total number of selected data is slightly more than the number of oracle data, and considering more data diversity potentially lead to better performance. We also discussed it in our future work part in our paper.
>
> > Why the other three data evaluation methods perform so poorly in Table 1?
>
> IFD assumes that all data in the dataset is of good quality where it selects informative data samples. It cannot effectively handle scenarios with poor-quality data, where IFD tends to select complex, difficult-to-learn noise patterns. PPL tends to select longer data samples. Both IFD and PPL are not well-equipped to deal with situations where malicious or intentionally corrupted data is present in the dataset. The dataInf method is only related to the optimal theta point and does not consider whether the validation loss actually decreases. This may result in selecting bad data that could increase the loss. In contrast, our approach can better handle settings with bad data.
>
> Additionally, DataInf makes several strong assumptions, which lead to more limitations such as:
> 1. the total approximation error is bounded by $O\left(\sum_{l=1}^L d_l^2\right)$, and the approximation error is tolerable only when $d$ is small.
> 2. The effectiveness of adapting DataInf to LoRA is limited to the number of learnable parameters is small, such as 1, 2, 4. The correlation coefficient of DataInf generally decreases as the rank increases.
>
> These limitations highlight the challenges in developing influence measurement functions and data selection, especially when dealing with diverse, heterogeneous data, whereas our proposed methods are good at broader scenarios.

---

> ### Author Response · Authors · 2024-08-13
>
> We have completed all the requested experiments and are pleased to share the results.
>
> Regarding generalizability beyond the medical domain, we conducted experiments using the FiQA dataset, which focuses on financial question answering.
>
> *Dataset:* We have four clients, each with 2000 training samples. We split the data into a training set (8000 samples in total), a validation set (also called the anchor set, 100 samples), and a test set (500 samples).
>
> *Evaluation metric:* Responses from the model fine-tuned on our dataset are rated by GPT-4 on a scale from 1 to 10, reflecting criteria including relevance, accuracy, and fluency. To address potential positional bias, we send our response along with the benchmark output to GPT-4 twice, with different orders. We then calculate the average of these scores as the final performance score.
>
> **Experiment I: Quality Heterogeneity**
>
> To evaluate our data selection method's effectiveness in scenarios with **heterogeneous data quality**, we simulated varying levels of data pollution across clients. Specifically, we randomly polluted 80%, 20%, 10%, and 50% of the training set for each of the four clients, respectively.
>
> $\textcolor{blue}{Table 1}$ presents the results of our model merging experiments. These findings demonstrate that our method significantly enhances data quality even when clients have differing proportions of low-quality data:
>
> $\textcolor{blue}{Table 1:}$
> |               | GPT-4 Scoring |
> | ------------- | ------------- |
> | Low-qual Data | 3.6495        |
> | Oracle        | 3.7480        |
> | **Ours (w/ linear merging)**      | 3.7755    |
> |**Ours (w/ weighted merging)**|	**3.8550**|
>
> We conducted an **ablation study** on our global threshold to further validate our approach. $\textcolor{blue}{Table 2}$ illustrates the advantage of using a global threshold determined by our anchor data for data selection in this heterogeneous setting, compared to selection based on average ratio or pre-determined scores. These results demonstrate that our approach successfully balances the identification of positive cases with the minimization of false positives, offering a robust and superior solution.
>
> $\textcolor{blue}{Table 2:}$
> |                            | Precision  | Recall     | F1 Score   | Accuracy   |
> | -------------------------- | ---------- | ---------- | ---------- | ---------- |
> | select by ratio            | 79.17%     | 79.17%     | 79.17%     | 75%        |
> | select by static score     | 92.77%	| 99.13%	| 95.84%	| 95.00%|
> | select by global threshold (**Ours**) | **97.44%** | **99.38%** | **98.39%** | **97.91%** |
>
> Additionally, we performed a **qualitative analysis** by manually comparing the outputs generated by models fine-tuned on our selected high-quality data versus the original low-quality data. This comparison ($\textcolor{blue}{Table 3}$) provided insights into the tangible improvements in model performance and output quality.
>
> $\textcolor{blue}{Table 3:}$
> | Questions | Model Response  (w/ data selection) | Model Response (w/ our data selection) |
> | ----------------- | ------------- | --------- |
> | How does a 2 year treasury note work? | It is a 2 year treasury note.  | Treasury notes are a short-term government security that matures in two years. They are issued by the U.S. Treasury Department to help finance the nation's deficit. |0.505|
> |What causes a stock to drop in price?|Investors are afraid to buy stocks because they are afraid the stock will drop in price. |If the stock price is falling, it's because there's a lot of supply, but not enough demand. If there's a lot of demand, but not enough supply, the price will rise. |

---

> ### Author Response · Authors · 2024-08-13
>
> **Experiment II: Varying Levels of Low-Quality Data**
>
> To evaluate the robustness of our data selection method under different data quality conditions, we conducted a series of experiments with varying proportions of low-quality data. We maintained a consistent proportion of low-quality data across all clients for each experiment, ranging from 0% to 100%, including 20%, 50%, and 80% pollution levels.
>
> $\textcolor{blue}{Table 4}$ presents the performance of models trained with and without our data selection method across these different proportions. The results demonstrate that our method effectively enhances data quality across all scenarios with **GPT-4 scoring**. And in terms of **accuracy** of the data selection, our method consistently selected over 99% of the high-quality data across different proportions of low-quality data.
>
> $\textcolor{blue}{Table 4:}$
> |                        | GPT-4 Scoring Performancew/o data selection | GPT-4 Scoring Performancew/ our data selection | Accuracy | Global Threshold (score) |
> | ---------------------- | ------------------------------------------- | ---------------------------------------------- | -------- | ------------------------ |
> | 0% bad data (raw data) | 3.7400                                      | ———                                            | ———      | ———                      |
> | 20% bad data           | 3.4735                                      | 3.6270                                         | 99.3125% | 0.0033                   |
> | 50% bad data           | 2.9625                                      | 3.7175                                         | 99.7250% | 0.0017                   |
> | 80% bad data           | 2.9155                                      | 3.8335                                         | 99.9375% | 0.0012                   |
> | 100% bad data          | 2.2860                                      | ———                                            | ———      | ———                      |
>
> To understand the adaptability of our global threshold, we analyzed how the global threshold changes with different proportions of low-quality data. $\textcolor{blue}{Table 4}$ illustrates that our **global threshold adjusts across varying levels of data quality**.
>
> -------
>
> > Limitation: The author's work is similar to federated learning. Should it be compared with the recent federated learning?
>
> Our work has covered federated learning as one of the collaborative training paradigms, and our evaluation was conducted in a federated learning setting. For a detailed discussion of empirical results, please refer to [our response to Reviewer ngAe regarding W2 & L1](https://openreview.net/forum?id=OU1uqd1vyw&noteId=rALL4TsjnZ). If you're referring to specific recent developments in federated learning that warrant additional comparison, we'd appreciate more details and elaboration.
>
> ----
>
> To summarize, these results collectively demonstrate the robustness and effectiveness of our method across a wide spectrum of data quality scenarios, reinforcing its potential for real-world applications.

---

> > ### Author Response · Authors · 2024-08-14
> > **Looking forward to your responses and further discussion!**
> >
> > We would like to thank you once again for your time and effort in providing feedback and comments. As there are less than 12 hours remaining for further discussion, we would greatly appreciate if you could let us know of any remaining concerns or points of confusion. We would be more than glad to discuss, address, and resolve them with you!

---

> ### Author Response · Authors · 2024-08-14
> **W4 & Q4 Further experimental results**
>
> > Why the other three data evaluation methods perform so poorly in Table 1?
>
> We have compared different data scoring methods for their accuracy in selecting high-quality data out of the mixed-quality datasets in the FiQA Experiment I setting mentioned above. Our method demonstrates significant advantages as follows.
>
> |          | Accuracy |
> | -------- | -------- |
> | PPL      | 58.44% |
> | IFD      | 58.90% |
> | **Ours** | **97.91%**   |

---

### Official Review · Reviewer_ngAe · 2024-07-13

**Soundness:** 2
**Presentation:** 2
**Contribution:** 2
**Rating:** 4
**Confidence:** 4

**Summary:**

This work proposes a method for finetuning LLMs in federated or collaborative training scenarios that selects the most informative examples for each client to train on such that their local parameter updates are likely to improve the global, merged model's performance on a public (shared) test set. The proposed two phase solution first scores local samples by computing the inner products between a gradient computed on a training sample and a gradient computed on a limited number of shared test samples, and only performs the next episode of local training on points that are above a specific quality threshold determined and distributed by the server in the previous round. Then, the server combines the parameter updates sent from each client, and using the new global parameters, iterates over the same shared anchor set to compute a new quality threshold to send back to the clients. By iterating this approach, in some settings they achieve performance improvements on medical domain tasks as compared to existing data valuation methods and baselines of training on curated subsets or all available data, without federation.

**Strengths:**

- The diagrams are well done and help the reader understand the method in an intuitive manner.

- The method is principled, and relatively simple to implement requiring only gradients (rather than higher order computations), it is grounded well in prior works on data valuation and active learning, and is amenable to efficient implementations when using adapter training methods as demonstrated by their LoRA based experiments.

- The ablations chosen are interesting, particularly the effect of layer selection for the gradient comparisons.

**Weaknesses:**

- The precise ordering of the algorithmic steps is not clear. Section 3.1-3.3 would greatly benefit from an Algorithm definition that clearly defines the order of operations regarding what gradients are computed on what data using what version of the parameters throughout the training process. It is hard to interpret the experimental results due to the lack of clarity in the section that introduces them.

- Unless the reviewer is misunderstanding something about the evaluation, Table 1, 2, and 3 are all misleadingly bolded. The row label "Ours" is sufficient to tell that the row indicates the proposed method, however, "Ours" is bolded in all three tables, and all columns, regardless of whether or not the method outperforms other valuation criteria, or baseline training settings. On a quick glance this suggests the method offers uniform improvement in all settings, but this is not the case. Please only bold the best score in each column and note the convention in the captions.

- The ablations (while appropriately chosen) are not convincing. The differences reported in 4.3 are likely not statistically significant, and in 4.4, the "hit rate" is not defined, which hurts because this part is of technical interest. Section 4.5 is confusing and the final sentence claims that the anchor score leads to the lowest proportion of low quality data, but this is exactly the opposite of Fig 4(c)?

**Questions:**

- It seems like the table and figure reference numbering might be off in a few places, please check the \ref commands.

- I would like to better understand the dataset construction process. There are 3 raw sources listed, but what is the exact sampling used to build the training dataset? How are the n=20 client local datasets selected (are they uniformly sampled from the full training or grouped in some way?) Also, critically, how are the test evaluation samples (the GPT4 evaluation 200) sampled, and what is the relation to the _anchor_ data? Are the anchor data held out Medalpaca, just like the test set?

- If the above description is somewhat correct, followup question is what are the statistics for which samples were selected in each round by each client as "above the quality threshold"... were these locally filtered subsets more often than not the medalpaca training questions? (and when running the other baseline scores, was this not true?)

My current hypothesis is that this grad x grad product is a similarity score between the local training data and the global test/anchor samples. Thus, it biases each client to train on the test-like data, and not the other training data.

**Limitations:**

- The primary limitation is in the fact that the method does not improve over the baselines or other scores in many settings evaluated. In Table 2 and 3, while "Ours" beats the low quality baseline, it doesn't beat the oracle or raw baselines often.
- Similarly in Table 1, "Ours" only beats the other quality criteria in the GPT4 based evaluation. Depending on the clarification about the construction of the anchor set, I worry that the method only works well when the test set and the anchor set are closely related which limits the real world applicability of the method.

I would be happy to engage with the authors to help improve the explication of the method and the presentation of the results, but my concerns with the strength of the empiricals are not insignificant. Clarifications would need to demonstrate a misunderstanding on my part of the evaluation setup or the results themselves in order for my score to improve.

---

> ### Author Rebuttal · Authors · 2024-08-07
>
> Thanks so much for your valuable comments and feedback.
>
> **W1**:
> > Detailed algorithm definition
>
> Thank you for your valuable suggestions! Following the reviewer's constructive feedback, we have included our pseudo-code algorithm in the PDF.
>
> **W2 & L1**:
>
> > Evaluation
> We appreciate your questions on the bolded numbers! We have fixed them all in our uploaded PDF.
>
> To further clarify our evaluation setup and clear up misunderstandings:
> Our preliminary results (Table 1 and Table 2 in the uploaded PDF) show severe performance degradation when mixing our constructed bad data into the raw data. This demonstrates the large impact of low-quality data in collaborative settings, which motivates the importance and pressing challenges for data selection methods in such settings. Our objective is to achieve better performance compared to the low-quality baseline, not the raw data performance. Oracle serves as the theoretical upper bound.
>
> To further highlight our empirical results:
> 1. For both pre-trained models and tasks, with other settings remaining the same, model merging performs better than federated learning. This indicates that loose communication between the local model and the server, compared to frequent communication, might lead to better generalization.
> 2. The performance boost with selected data in the federated setting is larger than in the model merging setting. This might be because during federated learning, we calculate the data score based on the global model (instead of the local model in the model merging setting) at each timestamp, which can better trace and regularize the training trajectory to the optimal location.
> 3. In both federated and model merging settings, our data selection can achieve over 96% and over 91% of the theoretical upper bound performance, respectively.
> 4. Our method outperforms the other centralized data selection baselines under the GPT4 Scoring metrics. Compared to the other methods which cause severe forgetting during instruction tuning, the performance of our method on the Knowledge-based benchmark remains within an acceptable range. This shows that our methods are able to improve domain-specific tasks without forgetting knowledge injected during pretraining.
>
> **Q1**: We appreciate your detailed suggestions on the table and figure reference numbering! We have fixed them all and will update in our final version.
>
> **Q2**:
> We split the whole dataset into the training set, the validation set (and anchor data), and the test set. They are all from the same data sources. The anchor data/validation set is in the same distribution as the training data, which is essential to ensure that the measurement of high-quality data aligns both during the selection (validation set and anchor set) and in the evaluation (test set).
>
> > Dataset construction process:
>
> For the Medical QA task: We use 16k samples in total, with 8k samples randomly sampled from PMC-LLama and Medalpaca-flashcards each. We uniformly partition the total samples into 20 clients in this task.
> For the Multilingual QA task: We use 6312 samples randomly sampled from MMedBench and 1052 samples per language for each of 6 clients. (Domain Heterogeneity setting)
> For our newly added Financial QA task: We randomly sampled 2000 data samples for each of 4 clients from the FiQA dataset, and polluted each of them with low-quality data ratios of 80%, 20%, 10%, and 50% respectively. (Quality Heterogeneity setting)
>
> We provided examples of raw data and low-quality data with their scores calculated by our proposed method in our uploaded PDF.
> > Test set:
>
> Test data are sampled from the same data sources. For open-ended GPT4 evaluation, the test samples are randomly sampled from Medalpaca-flashcards and MMedBench for the medical QA task and multilingual QA task, respectively.
> For knowledge-based evaluation, the test samples are multiple-choice tasks, which use accuracy as the main metric. This additional evaluation is to see if fine-tuning will degrade or forget the knowledge obtained from pretraining. Following convention, we adopt three prominent medical benchmarks for evaluation.
> > Anchor data and validation data:
>
> We followed the convention of existing data selection/valuation methods, which requires a clean validation set drawn from the target distribution. This setup has been adopted and proven to work well in many previous data selection methods, including Data Shapley and Influence Function, etc.[1] It doesn't limit the real-world applicability of the method, because there are always a few public high-quality data available; otherwise, we couldn't clearly define what high-quality data looks like.
>
> [1] LAVA: Data Valuation without Pre-Specified Learning Algorithms, ICLR 2023.

---

> > ### Comment · Reviewer_ngAe · 2024-08-11
> > **Response to Rebuttal**
> >
> > I appreciate the author's inclusion of the algorithm pseudocode. I have some followup questions on the order of operations.
> > 1) Is the new global threshold computed _before_ aggregating the client models from the previous round? This seems like introduces a lag, or gap between choosing the new threshold based on the state of the meta-model from the prior iteration, not the state of the model after incorporating the most recent client updates.
> > 2) Is there supposed to be a "for each checkpoint t in T" loop in the "On the Server" section? Section 3.2 "Step One" is the section in the manuscript that describes checkpoints being used, so it seems like this only applies in the local phase. Maybe I still don't understand the algorithm fully though.
> >
> > The clarifications provided are encouraging. Allow me to reiterate them to confirm that my new understanding of the key points of the work.
> >
> > A) (preamble in rebuttal) You create testbed scenario where clients have training datasets with some significant corruption/low quality data within them. You first show the impact of your intervention by comparing the results training on the raw data versus your corrupted data, and also include a baseline of training on just the remaining clean data, a reasonable upper bound on expected performance of an optimal filtering method. (This is made more clear as either s separate table as you show in the supplement, or by more clearly referring to "Raw Data", "Low-qual Data", and  "Oracle" as baseline measurements, or something other than "methods" because this is slightly confusing.
> >
> > B) (point 4. in your rebuttal) You compare your data selection algorithm method to others in Table 1 in the original draft (Table 4 in the supplement) and show that it performs better than PPL, IDF, and DataInf in 2 out of 4 scenarios with the other strongest method being PPL. Please bold the strongest method in each column, even when it is not your own. Using the Table 1 in the manuscript (the horizontal line separating baselines from selection methods is good), this means bolding Ours in Mistral GPT-4 column, bolding PPL in Mistral Knowledge column, boldingOurs in Llama GPT-4 column, and bolding PPL in Llama Knowledge column. The caption should say "the best performing selection method (row) in the lower subtable is bolded for each benchmark evaluation (column)."
> >
> > C) (points 1,2,3 in rebuttal) On Federated versus Model Merging... Can you explain how these are different with respect to your newly formalized algorithm in the supplement? I am not able to fully understand the reiterations of results in the rebuttal that discuss the differences because the draft does not make precise how the federated update or the model merging happens in context of the data selection process. (The model merging step, is now clarified in the supplement as a weighted average of the sent parameters $\theta'_k$ from clients. How would the federated update differ?) Point 2. in particular speculates that this is because of a key difference between how the data valuation scores are computed in the two settings - this is completely missing in the manuscript?

---

> > > ### Comment · Reviewer_ngAe · 2024-08-11
> > > **Response to Rebuttal (contd)**
> > >
> > > Finally, the data clarifications are _much_ appreciated and are critical to incorporate into the updated draft. It is important to clearly state the distributional relationship between the client training data, the client validation samples, the anchor data, and the final test data. (Correct me if I am still wrong but) In light of this new clarity, it seems that a simple explanation for how this method works is that the validation sets for each client are all known to be clean, and drawn from the test distribution. Therefore in expectation grad x grad products between local training samples and local validation samples will either be large when the training sample is clean, and small when the sample is corrupted. Critically though, this distinction will be relative, and this is a hypothesis for why a contextual threshold based on the current loss of the server model on the anchor data must be recomputed throughout the overall training process.
> > >
> > > Following this train of thought, was this need for a dynamic threshold ablated? What about the threshold update frequency? Would quantized threshold updates help or hurt the method? In general, was any analysis of the trajectory/statistics the score thresholds from the training runs performed? This would be a nice visual in an updated draft. Something like iterations on the x-axis, and the value of the prescribed global threshold on the y-axis. It could also show the fraction of client data that was above the current threshold at that step, and therefore trained on. Further/alternately, it could quantify the precision and recall of the local filtering - how many of the local samples for a client were correctly identified as clean (by passing the threshold) and how many of the corrupted samples were identified (by being omitted from the training round). It would be illuminating to see whether this threshold and the selections it causes have a stable or asymptotic dynamics or whether they are noisy throughout training.
> > >
> > > I note that the rebuttal did not address weakness 3 in my original review. I believe that these ablations are interesting, but they are not clearly presented so I would be unable to summarize what we are supposed to learn from them in the manuscript's current form.
> > >
> > > I am happy to bump my score up to a 4, but am unable to move it into acceptance territory at this time.

---

> ### Author Response · Authors · 2024-08-13
>
> **Evaluation:**
>
> Thank you for your suggestions on presenting the experimental results. Since we cannot directly modify the original PDF at this stage, we will implement the following changes in the updated version of the paper:
>
> * We will bold the best-performing selection method in each column of Table 1 (Table 4 in the supplement), as you suggested. This includes bolding "Ours" in the Mistral GPT-4 and Llama GPT-4 columns, and "PPL" in the Mistral Knowledge and Llama Knowledge columns.
>
> * We will update the table caption to read: "The best performing selection method (row) in the lower subtable is bolded for each benchmark evaluation (column)."
>
> If you have any further questions or concerns about the experimental setup or results, we would be more than glad to continue the discussion on OpenReview.
>
> -----
>
>
> **Threshold:**
>
> We do not use dynamic thresholds, nor is there any description related to dynamic thresholds in our submitted paper. We appreciate the opportunity to clarify this misunderstanding and address the confusion: The threshold is calculated only once throughout the entire pipeline. For details and proof on determining the global threshold, please refer to [our response to R1](https://openreview.net/forum?id=OU1uqd1vyw&noteId=xKx1wdiFS3).
>
> To clarify our method, here's a detailed breakdown of the data selection pipeline in collaborative fine-tuning, summerizing from Section 3.1 in our original paper and the pseudocode in our manuscript:
>
> * Stage 1 (On each client): Local fine-tuning with low-quality data; save model checkpoints.
> * Stage 2 (On each client): Calculate gradients, compute scores for each training sample, send scores to the server.
> * Stage 3 (On server): Calculate gradients, compute scores of anchor data, determine the global threshold using anchor data scores and client scores.
> * Stage 4 (On each client): Select data with scores not lower than the global threshold.
> * Stage 5 (On each client): Local fine-tuning with high-quality data, then send model parameters to the server.
> * Stage 6 (On server): Merge client model parameters to obtain the final global model.
>
> Your understanding that *"in expectation, grad x grad products between local training samples and local validation samples will be large for clean samples and small for corrupted samples"* is correct! However, we want to clarify that we do not use "a contextual threshold based on the current loss." As mentioned earlier, the threshold is calculated only once.
>
> -----
>
> **Federated versus Model Merging**
>
> We integrate both federated learning and model merging into a unified algorithmic framework because they fundamentally involve local training followed by model aggregation in the parameter space. Since weights in neural networks are updated through optimization algorithms based on accumulated gradients, our gradient-based method influences the model's weight space through selective gradient use. This underlying intuition allows our method to be applicable to both collaborative learning scenarios.
>
> The only difference between federated and model merging algorithms is: In the federated setting, there's periodic communication between the server (global model, or $\theta\_{merge}$ in our paper) and clients (local models). Clients perform local training between communication rounds. At each round, clients send their current model parameters to the server, which aggregates them to form the current global model, then sends it back to all clients. After each communication round, server and client models are aligned. In contrast, in the model merging setting, there's no periodic communication during local training; local models train independently throughout the entire process. Parameters are sent to the server only once, at the end of training, to form the final global model.
>
> To clarify **Point 2** further: In the federated setting, the checkpoints we save for each local client are based on the global model updated from the last model aggregation on the server, which typically occurs after a fixed number of local training epochs or iterations. This process incorporates information from other clients' models. Crucially, at the end of each communication round, the model parameters on the server and all participated clients are synchronized. This ensures that all participating clients start the next round from the same point, incorporating collective knowledge. This synchronization can help our data selection method better trace and regularize the training trajectory toward the optimal location.
>
> We don't calculate gradients or scores during the FL process itself; instead, we compute the scores only after the entire training is complete.

---

> ### Author Response · Authors · 2024-08-13
> **(contd)**
>
> (contd)
>
> We express the relationship and differences between federated learning and model merging using mathematical formulas as follows:
>
>
> **Model merging:**
>
> let $f_\theta \in \mathcal{F}$ denote the language model and $D\_k \in \mathcal{D}$ denote the training dataset on client $k$. Given the training datasets $D_k$, we can define a model merging operator $\mathcal{M}\_K( \cdot ; D_k , k \in K=\{1, \cdots, n\}): \mathcal{F} \rightarrow \mathcal{F}$.
>
> The model merging process can be expressed as
> $$
> f\_{merging}=\mathcal{M}\_K(f)
> $$
>
>
> **Federated Learning:**
>
> Based on the notation of model merging, the federated averaging process can be expressed as
>
> $$
> f\_{fed}= (\prod\_{t=1}^T \mathcal{M}\_{S\_t(K)})(f)
> $$
>
> where $T$ is the round number. $S_t(K)$ is the index set of selected datasets to train on each client at round $t$.
>
> ------
>
> **Ablation Studies:**
>
> Thank you for giving us the opportunity to clarify our approach!
>
> 1. Merging Techniques (Section 4.3): Figure 4(a) in our paper demonstrates that different weighted merging or aggregation techniques lead to varying performance. Notably, the performance of our data selection method with the Linear Merging technique doesn't even reach the performance of low-quality data with TIES merging technique, highlighting the **significant impact of weighted merging techniques** on overall performance. To make the comparison more clear, we presented it in the following $\textcolor{blue}{Table 1}$:
>
> $\textcolor{blue}{Table 1:}$
>
> |                              | GPT-4 Scoring Performance of Model Merging |
> | ---------------------------- | --------------------------------------- |
> | Raw Data                     | 0.505                                   |
> | Low-qual Data                | 0.485                                   |
> | Oracle                       | 0.490                                   |
> | **Ours (w/ linear merging)** | 0.470                                   |
> | **Ours (w/ TIES merging)**   | **0.487**                               |
>
> 2. Hit Rate and Low-Quality Data Selection (Sections 4.4 and 4.5):
> We apologize for the confusion. The "hit rate" in 4.4 and the "number of selected low-quality data" in 4.5 refer to the same metric: the proportion of bad data being filtered out. We presented this in two forms (percentage and data size) to provide different perspectives. A higher value indicates more effective filtering of bad data by our algorithm.
>
> 3. Additional Experiments: (Since the start of the rebuttal period, we have been conducting additional experiments to perform ablation studies. We now have comprehensive results that we believe are both convincing and robust. We are pleased to report these findings as follows)
>
> To provide a more comprehensive ablation study, in addition to the experiments in the "domain heterogeneity" setting shown above, we conducted additional experiments in a "quality heterogeneity" setting using the FiQA dataset, which focuses on financial question answering. We randomly polluted the training sets of four clients with varying degrees of low-quality data: 80%, 20%, 10%, and 50%, respectively. The following $\textcolor{blue}{Table 2}$ illustrates the advantage of using a global threshold determined by our anchor data for data selection:
>
> $\textcolor{blue}{Table 2:}$
> |                            | Precision  | Recall     | F1 Score   | Accuracy   |
> | -------------------------- | ---------- | ---------- | ---------- | ---------- |
> | select by ratio            | 79.17%     | 79.17%     | 79.17%     | 75.00%        |
> | select by hardcode score     | 92.77%     | 99.13%    | 95.84%     | 95.00%        |
> | select by global threshold (**Ours**) | **97.44%** | **99.38%** | **98.39%** | **97.91%** |
>
> These results demonstrate that our approach successfully balances the identification of positive cases with the minimization of false positives, offering a robust and superior solution. We have also conducted additional ablation studies on the global threshold, which can be found in Figure 2 of the updated manuscript.
>
> Furthermore, we conducted the ablation study of different merging techniques on the FiQA dataset, demonstrating the importance of weighted merging, shown in $\textcolor{blue}{Table 3}$. We use pairwise evaluation here, where responses from the model fine-tuned on our dataset are rated by GPT-4 on a scale from 1 to 10, reflecting criteria including relevance, accuracy, and fluency.
>
> $\textcolor{blue}{Table 3:}$
> |                              | GPT-4 Scoring |
> | ---------------------------- | --------------------------------------- |
> | Low-qual Data                | 3.650                                   |
> |Oracle| 3.748|
> | **Ours (w/ linear merging)**   | 3.776                               |
> | **Ours (w/ weighted merging)** | **3.855**                                  |

---

> > ### Author Response · Authors · 2024-08-14
> > **Looking forward to your responses and further discussion!**
> >
> > We would like to thank you once again for your time and effort in providing feedback and comments. As there are less than 12 hours remaining for further discussion, we would greatly appreciate it if you could let us know of any remaining concerns or points of confusion. We would be more than glad to discuss, address, and resolve them with you!

---

### Official Review · Reviewer_igLM · 2024-07-13

**Soundness:** 3
**Presentation:** 3
**Contribution:** 3
**Rating:** 7
**Confidence:** 3

**Summary:**

This paper proposes a novel approach for data quality control in collaborative fine-tuning of large language models (LLMs), particularly in settings where data cannot be directly shared between different silos due to privacy concerns. The authors introduce a method that scores training samples based on tracing gradients of low-rank adapters, filters out low-quality data locally, and then adaptively merges model parameters. They evaluate their approach on medical and multilingual question-answering datasets, demonstrating significant performance improvements compared to baselines.

**Strengths:**

1. The authors have presented their ideas in a well-organized and logical structure, making the paper easy to follow.
2. The paper addresses an important problem in collaborative LLM training, where data quality control is challenging due to privacy constraints. The proposed method shows promising results and could have significant implications for improving the performance of collaboratively trained language models across various clients.
3. The method leverages training dynamics, providing a more fine-grained and comprehensive approach to data quality measurement compared to traditional methods which only looked at the last model checkpoint.
4. The use of LoRA and layer selection in the method of tracing gradients cleverly makes it particularly well-suited for large language models, addressing the computational challenges often associated with training such models.

**Weaknesses:**

1. Although the methodology is thoroughly described, the authors could improve the clarity of Figure 3 by providing a more intuitive explanation of how training dynamics enhance collaborative training processes.
2. The paper contains some grammatical errors and typos. For example, in Table 3, "Performance of Data quality" should be "Performance of data quality", and on line 290, "the data influenced calculated via the last layer weights prone to a cancellation effect" should be corrected to "the data influence calculated via the last layer weights is prone to a cancellation effect".

**Questions:**

Please refer to weaknesses part

**Limitations:**

yes

---

> ### Author Rebuttal · Authors · 2024-08-07
>
> We thank the reviewer for the insightful and positive feedback!
>
> **W1 ans Q1**:
>
> > Although the methodology is thoroughly described, the authors could improve the clarity of Figure 3 by providing a more intuitive explanation of how training dynamics enhance collaborative training processes.
>
> Figure 3 demonstrates an optimized training approach for collaborative learning of multiple models. By selecting high-quality training data for each local model, we are selecting gradients that positively impact loss trajectories. These trimmed gradients are accumulated, leading to an improved position in the weight space. By considering the interference during our data selection (gradient selection) of $\Delta\theta'{1}$ and $\Delta\theta'{2}$, we reduce the interference of weight updates from different models. After parameter aggregation, the merged model $\Delta\theta_{merged}$ can be improved to an enhanced position in the weight space represented by $\Delta\theta_{targeted}$.
>
> **W2 and Q2**:
>
> > The paper contains some grammatical errors and typos. For example, in Table 3, "Performance of Data quality" should be "Performance of data quality", and on line 290, "the data influenced calculated via the last layer weights prone to a cancellation effect" should be corrected to "the data influence calculated via the last layer weights is prone to a cancellation effect".
>
> We appreciate your detailed suggestions on the typos! We have fixed them all and will update them in our final version.

---

> ### Comment · Reviewer_igLM · 2024-08-12
>
> Thank the author for providing further information and contributing great work to the community! I have read other reviewers' comments and the rebuttal information. I would like to remain my rating as 'Accept.'

---

> > ### Author Response · Authors · 2024-08-14
> >
> > We are grateful for the reviewer's endorsement! We would also like to highlight that we have additional results on FiQA with different settings that all support our claims and conclusions, further strengthening our contribution. We plan to add all of these results to the final version of our paper and the appendix. For more details, please refer to the global comment and our response to Reviewer V7za.

---

### Official Review · Reviewer_Xfnf · 2024-07-22

**Soundness:** 2
**Presentation:** 2
**Contribution:** 2
**Rating:** 5
**Confidence:** 5

**Summary:**

This paper proposes a data quality control technique for the collaborative training of large language models from filtered private heterogeneous data domains via a quality score function that tracks the gradient of each training sample. The proposed framework is tested in medical and multilingual settings to demonstrate its effectiveness.

**Strengths:**

- This paper studies an important issue in private data quality control for collaborative model development, especially for the applications of LLMs.
- The proposed gradient-based score function is straightforward and effective.

**Weaknesses:**

- The assumption of homogeneity for model architecture, usage of low-rank adaptation, and anchor data with global threshold limit the overall applicability of the proposed framework.
- The experimental settings/details and the ablation study are limited.

**Questions:**

- Why use a global threshold as the unified standard of data quality? How were the anchor data selected? Would the usage of anchor data (validation set) introduce an additional bias toward the overall data quality (some datasets could potentially be all excluded) and harm the generalization ability? It would be better to see the performance comparison between global and local thresholds.
- How is the per-sample gradient computed from Eq. (1) line 172? Could the authors provide a complexity analysis on this part?
- Line 196, “sparse nature of low-rank gradients” refers to the trainable weight matrices being low rank or the obtained per-sample gradient being sparse and low-rank.
- It would be great to show the ablation study of directly applying the proposed methods to raw data (supposed high-quality data), given the federated results reported in Table 2.
- Could the authors provide several examples of low-quality data samples as presented in Table 4?

**Limitations:**

See W1-2, Q1-2, 4

---

> ### Author Rebuttal · Authors · 2024-08-07
>
> **W1**
>
> > The assumption of homogeneity for model architecture
>
> We clarify that we follow the commonly agreed assumption of previous work on collaborative learning (federated learning, model merging) [1, 2, 3]: in order to aggregate the model in the parameter space, we should have the same model architecture or the same adapter. It's impossible to perform model aggregation if the models have different architectures. Exploring different model architectures may be beneficial for other applications but is not the focus of our paper; for example, multimodality.
>
> > Usage of low-rank adaptation
>
> We wanted to clarify that the setup and motivation for using collaborative fine-tuning is to gather knowledge in an efficient manner. We focus on situations where individual clients don't have enough data or compute resources to pre-train a large language model or even perform full parameter fine-tuning. Low-rank adaptation is a commonly used parameter-efficient fine-tuning method for such scenarios. Therefore, the usage of low-rank adaptation is not a limitation but a way to achieve broader applicability. It is also regarded as a strength from other reviewers' perspectives: ngAe and igLM.
>
> **W2**
>
> > The experimental settings/details and the ablation study are limited.
>
> We have provided more ablation studies and experimental details in the uploaded PDF. We would really appreciate hearing your feedback regarding the limitations in more detail!
>
> -------
> **Q1**
>
> > Global threshold as the unified standard, versus local thresholds:
>
> It is necessary to have a global threshold to deal with the heterogeneity of data quality. Since the ratio of local low-quality data is unknown, and the ratios for each client are different from each other, local thresholds are not able to adapt to different and diverse low-quality data ratio scenarios. We show the comparison between global and local thresholds in Figure 1 of our PDF. There, the local optimal threshold represents the optimal thresholds for each client and is unknown in practice. The global threshold can be close to the local optimal, while local thresholds are far from the optimal thresholds.
>
> > Anchor data selection and biases
>
> The anchor data is selected as a held-out high-quality dataset from the same data source as the test data. We show the ablation study of the anchor data selection and validation set selection in Figure 2 in our uploaded PDF, which demonstrates the robustness and unbiased selection in our experiments. In Figure 2 (Left), the global threshold remains stable and robust with different numbers of anchor data selected. In Figure 2 (Right), the box plot shows the data scores for 20 training samples randomly sampled from our training dataset over 5 different validation sets of equal size. Although the scores vary depending on the validation set used, the variance is within an acceptable range, ensuring that the order of their scores remains stable and robust. Thus, bias is not a concerning factor in our paper. We acknowledge that, in other cases, validation set selection will significantly affect the data selection process, and this is a common issue for all data selection work using validation sets.
>
>
> **Q2**
>
> Per-sample gradients are calculated for each training sample from the checkpoint saved during the model training.
>
> For SGD,
> $\boldsymbol{\theta}^{t+1}-\boldsymbol{\theta}^t=-\eta_t \nabla \ell\left(\boldsymbol{z} ; \boldsymbol{\theta}^t\right)$
>
> For Adam, $\boldsymbol{\theta}^{t+1}-\boldsymbol{\theta}^t=-\eta_t \mathcal{L}\left(\boldsymbol{z}, \boldsymbol{\theta}^t\right) $,  $\mathcal{L}\left(\boldsymbol{z}, \boldsymbol{\theta}^t\right) \triangleq \frac{\boldsymbol{m}^{t+1}}{\sqrt{\boldsymbol{v}^{t+1}}+\epsilon} $
>
> For AdamW, $\boldsymbol{\theta}^{t+1}-\boldsymbol{\theta}^t=-\eta_t \mathcal{L} \left(\boldsymbol{z}, \boldsymbol{\theta}^t\right) $, $\mathcal{L}\left(\boldsymbol{z}, \boldsymbol{\theta}^t\right) \triangleq \frac{\boldsymbol{m}^{t+1}}{\sqrt{\boldsymbol{v}^{t+1}}+\epsilon}{+\lambda \boldsymbol{\theta}^t}$
>
> - overall compute complexity, where $N$ is number of checkpoints, and $d$ is gradient dimension.
> $$
> \mathcal{O}\left(N \cdot |\mathcal{D}| \cdot\left|\mathcal{D}_{\text {val }}\right| \cdot d\right)
> $$
> - overall storage complexity:
> $$
> \mathcal{O}(|\mathcal{D}| \cdot N \cdot d + |D_{val}| \cdot N \cdot d)
> $$
>
> **Q3**
>
> We use LoRA to reduce the number of trainable parameters, which freezes the pre-trained weights and adds a low-rank adaptor to linear layers throughout the network. This means that, by nature, the trainable matrices are low rank. While our method is based on LoRA, it would also be very easy to adapt our proposed method to training where per-sample gradients are sparse and low-rank, for example, GaLore [4]. Our method is orthogonal to different training methods, and we believe this is a very promising direction for another future work.
>
> **Q4**
>
> > ablation study of directly applying the proposed methods to raw data (supposed high-quality data).
>
> As we clarified in our global response, raw data is not one of the baselines we are comparing our data selection method with, since the scope of our paper is to conduct data selection to filter out the low-quality data mixed in with the high-quality data. However, we agree it would be interesting to see how our methods perform on the dataset without low-quality data. We are currently running these experiments and will update the results once we have them.
>
> **Q5**
> > Examples of low-quality data samples
>
> Sure, we have provided the low-quality and high-quality data samples with their scores in our PDF for your kind review.
>
> [1] TIES-Merging: Resolving Interference When Merging Models, NeurIPS 2023.
>
> [2] Editing Models with Task Arithmetic, ICLR 2023.
>
> [3] Communication-Efficient Learning of Deep Networks From Decentralized Data, Artificial Intelligence and Statistics, 2017.
>
> [4] GaLore: Memory-Efficient LLM Training by Gradient Low-Rank Projection, ICML 2024.

---

> > ### Comment · Reviewer_Xfnf · 2024-08-12
> >
> > Thank the authors for the response and clarification. The added new results certainly help us understand the proposed framework.
> >
> > - Regarding W1, I wanted to highlight the potential use case for mixed fine-tuning with other PEFT methods and LoRA with different parameters (e.g. rank).
> > - Q1, I suggest that the authors incorporate the discussion of the necessity and justification for global thresholds and their implications. Also, the current conclusion is based on limited observation from the selected datasets.
> > - Q3 Here, do the authors use macro-batching to obtain the per-sample gradient? If that's the case, then I reserve the scalability concerns over the proposed framework.
> >
> > In recognizing the effort of the authors for rebuttal, I increase my score to 5.

---

> ### Author Response · Authors · 2024-08-13
>
> First of all, we would like to clarify the following definition and notations:
>
>
> **Definition 1.1** (Data Quality on Specific Domain $k$)**.** Given a model architecture $\theta$, a training configuration (optimizer, etc.), and a validation set $D_{val}$ in a specific domain $k$, the quality of training data $z$ is defined as follows: for $z\_1, z\_2 \in \mathcal{D}\_{train}$, if $\mathcal{L}\_{val}(\theta(z\_1), D\_{val}) < \mathcal{L}\_{val}(\theta(z\_2), D\_{val})$, then the quality of $z_1$ is considered higher than that of $z_2$. Here, $\mathcal{L}_{val}$ denotes the validation loss. In other words, the lower the validation loss, the higher the data quality.
>
>
> **Definition 1.2** (Data Quality on Collaborative Private Domains)**.** Given a model architecture $\theta$, a training configuration (optimizer, etc.), and a validation set $D\_{val}=\{\mathcal{D}^{(1)}\_{val}, \mathcal{D}^{(2)}\_{val}, \ldots, \mathcal{D}^{(K)}\_{val}\}$ for all $K$ tasks, the quality of training data $z$ is defined based on the validation loss of **the global model** $\theta_{merged}$ on $D_{val}$. Specifically, for $z\_1, z\_2 \in \\mathcal{D}^{(k)}\_{train}$, if $\mathcal{L}\_{val}(\theta\_{merged}(z\_1), D\_{val}) < \mathcal{L}{val}(\theta_{merged}(z\_2), D\_{val})$, then the quality of $z_1$ is considered higher than that of $z_2$. As in the single-domain case, lower validation loss indicates higher data quality.
>
> **Remarks** (Enhancing Data Quality on Collaborative Private Domains)**.** In the collaborative learning framework, the ratio and distribution of low-quality data are unknown a priori. Only the server has access to the global distribution of both high-quality and low-quality data, while individual users cannot infer the global distribution from their local distributions due to statistical heterogeneity. The server can infer the distribution of high-quality data from public anchor data. Our objective is to select data points that most significantly reduce the validation loss of the global model, rather than optimizing for each local model independently. It is important to note that the scope of this study is not considering new models joining during training or continual learning paradigms.
>
> --------
> Below are our responses to each follow-up question. We appreciate your insightful comments and the time and effort you've engaged in helping us clarify our proposed method. We look forward to continuing discussion with you on the OpenReview system if you have any remaining concerns, questions, or if you feel there's any misunderstanding on our part regarding the questions!
>
> **1. Mixed Model Architectures:**
>
> As per **Definition 1.1** above, data quality is specific to each model architecture. Different architectures naturally lead to the selection of different high-quality data points. Under the current definition of data quality, it is not a reasonable assumption to uniform data selection across models with varying configurations. For example, the data quality standard would differ significantly between LoRA models with rank $r=16$ and those with $r=4$.
>
> When it comes to individual models or groups of models from different domains with identical configurations, our proposed method is orthogonal and can be broadly applied. It is compatible with any model that allows gradient computation and can seamlessly integrate with state-of-the-art PEFT techniques such as AdaLoRA or Prompt Tuning.
>
> Given that current collaborative learning and distributed training scenarios predominantly focus on models with homogeneous architectures, we believe our algorithm has broad applicability. While we acknowledge the potential for mixed fine-tuning with various PEFT methods, our current focus is on providing a SOTA solution for the most common use cases in the field.

---

> ### Author Response · Authors · 2024-08-13
>
> **2. Global Threshold:**
>
> > I suggest that the authors incorporate the discussion of the necessity and justification for global thresholds and their implications
>
> According to **Definition 1.2** and **Remarks**, global thresholds are essential in our framework to effectively handle **domain heterogeneity** mentioned in our paper. As we highlight in our paper, local models can interfere with each other when merged in the weight space. Since our objective is to reduce the validation loss of the global model rather than local models, data selected by local thresholds don't necessarily translate to an optimal global model when merged. Global thresholds are necessary to coordinate different local distributions, thereby avoiding potential conflicts or interference during model merging or aggregation.
>
> >  The current conclusion is based on limited observation from the selected datasets.
>
> We have also conducted experiments on the FiQA dataset with different settings (see more details in our response to Reviewer V7za). They all demonstrate the same pattern, showing that the global threshold minimizes the sum of the distances to each local optimal threshold, while local thresholds cannot lead to global optimal solution. According to the instructions for the discussion period, we are not allowed to include new figures. However, we will incorporate figures showing the score distribution with different thresholds in the final version of our paper.
>
>
> In addition, we provide details and proof on how we determine the global threshold as follows:
>
> Suppose the score distribution of all data (including both good and bad data) is denoted by $F(x)$. The score distributions for good data and bad data are represented by $f(x)$ and $g(x)$ respectively. The relationship between these distributions can be expressed as:
> $$
> F(x)=\alpha*f(x)+(1-\alpha)*g(x)
> $$
> where $\alpha$ is the proportion of good data in the dataset.
>
> We can estimate $F(x)$ using the scores from all clients:
> $$
> \hat F(x) = \Sigma_{z_i \in D_{train}} kernel(x-S(z_i))
> $$
> Similarly, we estimate $f(x)$ using the scores from anchor data:
> $$
> \hat f(x) = \Sigma\_{z\_i^* \in D\_{anchor}} kernel(x-S(z^*\_i))
> $$
> Assuming $\alpha$ is known, which can be estimated from the quality of the global data, we can estimate the bad data distribution as follows:
> $$
> \hat g(x)=\frac{\hat F(x)-\alpha*\hat f(x)}{1-\alpha}
> $$
> The optimal global threshold $\tau^*$ can be obtained by :
> % $$
> % (1-\alpha)\hat g(\tau^*)=\alpha*\hat f(\tau^*)
> % $$
> % This implies that the proportion of good data above the threshold $\tau^*$ is maximized.
> $$
> \tau^*= argmax\{\frac{\int_{\tau}^{+\infty}\alpha\hat f(x) \, dx}{\int_{\tau}^{+\infty}(1-\alpha)\hat g(x) \, dx}+\lambda\int_{\tau}^{+\infty}\hat F(x) \, dx\}
> $$
> where $\lambda$ is a hyper-parameter that balance the data number and good data ratio.
>
> For a specific submodel $k$, the algorithm can be further refined. We denote the estimated good data distribution for a specific data source as $\hat{f}^{(k)}(x)$:
>
> $$
> \hat f^{(k)}(x) = \Sigma_{z_i^* \in D^{(k)}_{anchor}} kernel(x-S(z^*_i))
> $$
> The threshold for the specific submodel can then be determined by:
> % $$
> % \hat g(\tau^*)=\hat f^{(k)}(\tau^*)
> % $$
>
> $$
> \tau^*= argmax\{\frac{\int_{\tau}^{+\infty}\alpha\hat f^{(k)}(x) \, dx}{\int_{\tau}^{+\infty}(1-\alpha)\hat g(x) \, dx}+\lambda\int_{\tau}^{+\infty}\hat F(x) \, dx\}
> $$
>
> ------
>
> **3. Per-sample gradient calculation:**
>
> We appreciate the opportunity to clarify our approach to calculating sample gradients from a checkpoint:
> For a given checkpoint with parameters $\theta_t$, we compute the gradient of a single sample $z_i$ as follows:
> $$
> \nabla\ell\left(\boldsymbol{\theta}\_t;\boldsymbol{z}\_i\right) =\nabla\_{\theta_t} \text{LOSS}(f\_{\theta_t}(x\_i),y\_i)
> $$
> where $x_i$ and $y_i$ are the input and label of $z_i$, respectively.
> To enhance computational efficiency, we can focus on specific parts of $\theta$, such as individual layers or LoRA modules (e.g., q, k, v, o). For instance, if we concentrate solely on the gradients of parameters in layer 0, we adjust our gradient calculation as:
> $$
> \nabla\ell\left(\boldsymbol{\theta}\_t;\boldsymbol{z}\_i\right) =\nabla\_{\theta\_t^{layer0}} \text{LOSS}(f\_{\theta\_t}(x\_i),y\_i)
> $$
> It's important to note that this gradient computation process is based on a single checkpoint, with no parameter updates occurring throughout. This allows us to process each training data point in parallel, significantly improving computational efficiency.
>
>
> Regarding your concern about scalability, did you mean "micro-batching" here? In our scoring calculation step, we use a micro batch size of 1. Since there are no model updates during this process, scalability is not an issue.
>
> We'd be glad to open-source our code implementation for more details.

---

> > ### Comment · Reviewer_Xfnf · 2024-08-13
> >
> > Thanks for the response. As for point 3, it should be 'micro-batching.' And I understand that the computation of the per-sample gradient is based on a single checkpoint. But I'm not sure what the authors mean here by "allow us to process each training data point in parallel". Popular auto differentiation libraries such as PyTroch provide the aggregated gradient for a batch but not per-sample gradients. So, to get the per-sample gradient, micro-batching is generally used to run backpropagation one time per data sample, where the benefit of parallelism is lost. So, could the authors further elaborate on why this part's scalability is **not** an issue?

---

> ### Author Response · Authors · 2024-08-13
>
> We appreciate your insightful question regarding the scalability of our method!
>
> From an implementation perspective, per-sample gradient calculation is a widely encountered quantity in differential privacy and meta-learning. There are many tools we can use to compute per-sample gradients efficiently if needed, such as: (1) the Opacus library, which focuses on extending PyTorch for differential privacy, or (2) functorch, where composing *vmap* and *grad* provide significant speedup. In general, vectorization with *vmap* should be faster than running a function in a for-loop and competitive with manual batching.
>
> Moreover, it's important to clarify the context of our work: our focus is on collaborative fine-tuning scenarios where clients have limited resources and data, naturally constraining the scale of per-client data selection. This setting often doesn't involve extremely large datasets on individual clients. In multi-GPU setups, we can leverage parallel computation across GPUs to enhance efficiency.
>
> For cases involving clients with extremely large datasets or single GPU, in addition to using tools orthogonal to our proposed method like the Opacus library or functorch, we can also adapt our algorithm to address scalability requirements: We can compute a coarse-grained per-batch score, select all samples in batches exceeding our per-batch threshold, and apply per-sample gradient computation only to lower-scoring batches. This extended approach allows us to balance efficiency and precision in data selection, addressing potential scalability issues in large-scale scenarios from the algorithm perspective.

---

> ### Author Response · Authors · 2024-08-14
> **Looking forward to your feedback and further discussion!**
>
> We would like to thank you once again for your time and effort in providing insightful feedback and comments. As there are only a few hours remaining for further discussion, could you kindly let us know if you have any remaining concerns or points of confusion? We would be more than happy to discuss, address, and resolve them with you! If our responses have addressed your follow-up concerns, we would greatly appreciate your considering an increase in your rating to reflect these new improvements and clarifications, which we'll incorporate in our updated paper.
>
> Thank you for your continued engagement and support in strengthening our research!

---

> ### Author Response · Authors · 2024-08-14
> **Experimental results on raw data**
>
> **Q4**
> > ablation study of directly applying the proposed methods to raw data (supposed high-quality data).
>
> We have completed all the requested experiments and are pleased to share the results.
>
> Regarding generalizability beyond the medical domain, we conducted experiments using the FiQA dataset, which focuses on financial question answering. For other results, please refer to our discussion with Reviewer V7za.
>
> *Dataset*: We have four clients, each with 2000 training samples. We split the data into a training set (8000 samples in total), a validation set (also called the anchor set, 100 samples), and a test set (500 samples).
>
> *Evaluation metric*: Responses from the model fine-tuned on our dataset are rated by GPT-4 on a scale from 1 to 10, reflecting criteria including relevance, accuracy, and fluency. To address potential positional bias, we send our response along with the benchmark output to GPT-4 twice, with different orders. We then calculate the average of these scores as the final performance score.
>
> The table presents the performance of models trained with and without our data selection method across these different proportions. The results demonstrate that our method effectively enhances data quality across all scenarios with **GPT-4 scoring**. And in terms of **accuracy** of the data selection, our method consistently selected over 99% of the high-quality data across different proportions of low-quality data.
>
> |                        | GPT-4 Scoring Performancew/o data selection | GPT-4 Scoring Performancew/ our data selection | Accuracy | Global Threshold (score) |
> | ---------------------- | ------------------------------------------- | ---------------------------------------------- | -------- | ------------------------ |
> | 0% bad data (raw data) | 3.7400                                      | ———                                            | ———      | ———                      |
> | 20% bad data           | 3.4735                                      | 3.6270                                         | 99.3125% | 0.0033                   |
> | 50% bad data           | 2.9625                                      | 3.7175                                         | 99.7250% | 0.0017                   |
> | 80% bad data           | 2.9155                                      | 3.8335                                         | 99.9375% | 0.0012                   |
> | 100% bad data          | 2.2860                                      | ———                                            | ———      | ———                      |
>
> To understand the adaptability of our global threshold, we analyzed how the global threshold changes with different proportions of low-quality data. It illustrates that our **global threshold adjusts across varying levels of data quality**.

---

### Author Rebuttal · Authors · 2024-08-07

We thank the reviewers for their valuable feedback and appreciate the great efforts made by all reviewers, ACs, SACs, and PCs. We are grateful that the reviewers have multiple positive impressions of our work, including:
* *[Motivation]* **studies an important issue** (Xfnf, igLM)
* *[Method]* **straightforward and effective** (Xfnf) and **principled, and relatively simple to implement requiring only gradients (rather than higher order computations)** (ngAe) and **provides a general pipeline applicable to various models without the need for task-specific adjustments**(V7za)
* *[Results]* **promising results and could have significant implications** (igLM)
* *[Presentation]* **diagrams are well done and help the reader understand the method in an intuitive manner** (ngAe), **presented their ideas in a well-organized and logical structure, making the paper easy to follow** (igLM), and **the structure of the article is complete and easy to understand**(V7za).


Firstly, we would like to restate and clarify our assumptions for data selection in collaborative fine-tuning to help audiences better understand the scope of our work:
* We maintain the same setup as other previous works. To enable collaborative fine-tuning, each model shares the same architecture or the same LoRA adapter to adapt to different target tasks. To improve the efficiency of assigning utilities to each data point, we only calculate gradients for the parameter-efficient QLoRA modules.
* Evaluation Setup: For all methods, we use the same setup following previous literature. Our raw training set, hold-out validation set, and test set are IID. For the experimental setup, they are from the same data source. For the Medical QA task, we split the MedAlpaca dataset into raw training set, hold-out validation set, and test set. For the multilingual task, we split the MMedBench dataset into raw training set, validation set, and test set. For the low-quality training set, we pollute the raw training set by mixing data from other domains into it; more details are in the PDF. In our paper, we focus on in-domain evaluation and leave out-of-domain or cross-domain (where training set and test set are from different data sources) for future work.

Secondly, we wanted to highlight our contributions drawn from our method design and experimental results:
* To the best of our knowledge, we are the first to propose a data selection method for large language models in a collaborative setting, while previous work has mainly focused on traditional centralized settings. We bring up the insights to view federated learning and model merging within the same framework, incorporate different experimental setups and unify federated learning and model merging methods, making it universally applicable.
* Our method performs well on generation datasets and takes into account scenarios with bad data, while previous work has not considered downstream domain-specific generation tasks for large language models. Our method does not require repeated training.

We once again express our gratitude to all reviewers for their time and effort devoted to evaluating our work. We eagerly anticipate your further responses and hope for a favorable consideration of our revised manuscript.

---

> ### Author Response · Authors · 2024-08-14
> **Summary of follow-up questions and discussions**
>
> We would like to thank all the reviewers for their comments and discussions, and provide a summary of our responses. We will add all corresponding discussions and experimental results to our updated paper.
>
> **Background and Motivation:**
>
> - We have clarified our assumptions on collaborative settings, definition of data quality, and goals of data selection in the collaborative setting. (for Reviewer Xfnf)
> - We have validated the reasonableness of our low-quality data construction methods: a larger portion of low-quality data has resulted in lower validation loss. (Please refer to the $\textcolor{blue}{Table}$ showing validation loss at the bottom of this response.)
> - We have clarified the data sources of validation data and anchor data. (for Reviewer Xfnf and ngAe)
> - We have discussed the same nature of federated learning and model merging. We have expressed the relationship and differences between federated learning and model merging using mathematical formulas. (for Reviewer ngAe)
>
> **Methods:**
>
> - We have refined the notations for improved clarity and have provided more detailed pseudo-code for our method in the PDF file. (for Reviewers igLM, ngAe)
> - We have elaborated on the order of the whole procedure. (for Reviewer ngAe)
> - We have elaborated on how we determine the global threshold. (for Reviewer Xfnf)
> - We have discussed how we calculate the per-sample gradient (for SGD, Adam, and AdamW) and its complexity and scalability. (for Reviewer Xfnf)
> - We have offered additional explanation on our method's training dynamics. (for Reviewer igLM)
> - We have provided low-quality and high-quality data samples with their scores in our PDF. (for Reviewer Xfnf)
>
> **Experiments:** *We have spent considerable time and effort preparing these new results. We have more results that all support our claims and conclusions and we plan to add all of them to the paper or to the appendix.*
>
> - We have added experiments on a new dataset from a data-sensitive private domain - financial question answering. (for Reviewer V7za)
> - We have clarified the baselines we are comparing to in our PDF. (for Reviewer ngAe)
> - We have conducted evaluations on two challenging scenarios mentioned in the introduction of our paper: domain heterogeneity and quality heterogeneity.
> - We have elaborated on our ablation study on weighted merging techniques. (for Reviewer ngAe)
> - We have presented a comparison study on the global threshold determined by anchor data, comparing with other thresholds. (for Reviewers Xfnf, ngAe, and V7za)
> - We have provided further discussions of the experimental results, including a comparison of data selection between the federated setting (frequent communication) and model merging (loose communication) setting and the potential reasons for the results. (for Reviewer ngAe)
> - We have experimented with varying proportions of low-quality data to understand the method's effectiveness across different levels of data quality. (for Reviewer V7za)
> - We have compared our gradient-based method with other centralized data selection methods including PPL, IFD, and DataInf, and have shown our advantages. (for Reviewer V7za)
> - We have conducted an ablation study on the impact of using different numbers of anchor samples on the performance and reliability of the method. (for Reviewer V7za)
>
> We once again express our gratitude to all reviewers for their time and effort devoted to evaluating our work. We hope our discussion and summary have addressed your concerns, misunderstandings, or questions.
>
> --------
>
> $\textcolor{blue}{Table}$: A larger portion of low-quality data results in lower validation loss. After data selection, the validation loss becomes the same as that of the raw data, demonstrating the effectiveness of our low-quality data construction and data selection methods.
>
> |                        | Validation Loss | Validation Loss After Data Selection |
> | ---------------------- | --------------- | ------------------------------------ |
> | 0% bad data (raw data) | 1.77342         | —————                                |
> | 20% bad data           | 1.77893         | 1.77397                              |
> | 50% bad data           | 1.80144         | 1.77344                              |
> | 80% bad data           | 1.85195         | 1.77322                              |
> | 100% bad data          | 1.93056         | —————                                |

---

### Decision · Program_Chairs · 2024-09-25

**Decision:**

Accept (poster)

**Comment:**

The paper proposes a data selection method for large language models in a collaborative fine-tuning setting. The authors present a framework that unifies federated learning and model merging approaches, applicable without task-specific adjustments.

The proposed method scores training samples by tracing gradients, filters out low-quality data locally, and adaptively merges model parameters offering potential benefits over naive averaging as per FedAvg, model soups and BTM. The authors evaluate their approach on medical and multilingual question-answering datasets, demonstrating performance improvements compared to established baselines.

Reviews are mixed but leaning positive. Most negative aspects have been addressed in the response. Some reviewers noted that the precise ordering of the algorithmic steps was not clear in the original submission. The authors have responded by including pseudo-code in the rebuttal supplemental materialks. There were also concerns regarding the presentation of results, particularly the bolding of values in tables, which may mislead readers into thinking that the method uniformly outperforms all baselines. The authors have provided justification in their rebuttal which should also be added in the final manuscript.

The method’s effectiveness seems to hinge on the relationship between the anchor set and the test set, raising concerns about its generalisability to other settings. The authors have clarified their dataset construction and evaluation processes, but the method’s applicability in more diverse, real-world scenarios remains a potential limitation.